# Bridging Explicit and Implicit Deep Generative Models via Neural Stein Estimators

**Qitian Wu**[1*] **Rui Gao**[2*] **Hongyuan Zha**[1,3]
[1]Department of Computer Science and Engineering,
MoE Key Lab of Artificial Intelligence, AI Institute, Shanghai Jiao Tong University
[2]University of Texas at Austin
[3]School of Data Science, Shenzhen Institute of Artificial Intelligence and Robotics for Society,
The Chinese University of Hong Kong, Shenzhen
echo740@sjtu.edu.cn, rui.gao@mccombs.utexas.edu, zhahy@cuhk.edu.cn

## Abstract

There are two types of deep generative models: explicit and implicit. The former defines an explicit density form that allows likelihood inference; while the latter targets a flexible transformation from random noise to generated samples. While the two classes of generative models have shown great power in many applications, both of them, when used alone, suffer from respective limitations and drawbacks. To take full advantages of both models and enable mutual compensation, we propose a novel joint training framework that bridges an explicit (unnormalized) density estimator and an implicit sample generator via Stein discrepancy. We show that our method 1) induces novel mutual regularization via kernel Sobolev norm penalization and Moreau-Yosida regularization, and 2) stabilizes the training dynamics. Empirically, we demonstrate that proposed method can facilitate the density estimator to more accurately identify data modes and guide the generator to output higher-quality samples, comparing with training a single counterpart. The new approach also shows promising results when the training samples are contaminated or limited.

## 1 Introduction

Deep generative model, as a powerful unsupervised framework for learning the distribution of high-dimensional multi-modal data, has been extensively studied in recent literature. Typically, there are two types of generative models: explicit and implicit. Explicit models define a density function of the distribution [35, 51, 42], while implicit models learn a mapping that generates samples by transforming an easy-to-sample random variable [15, 39, 2, 4].

Both models have their own power and limitations. The density form in explicit models endows them with convenience to characterize data distribution and infer the sample likelihood. However, the unknown normalizing constant often causes computational intractability. On the other hand, implicit models including generative adversarial networks (GANs) can directly generate vivid samples in various application domains including images, natural languages, graphs, etc. Nevertheless, one important challenge is to design a training algorithm that do not suffer from instability and mode collapse. In view of this, it is natural to build a unified framework that takes full advantages of the two models and encourages them to compensate for each other.

Intuitively, an explicit density estimator and a flexible implicit sampler could help each other's training given effective information sharing. On the one hand, the density estimation given by explicit

---

[*]Part of the work was done when the two authors were visiting The Chinese University of Hong Kong, Shenzhen.

35th Conference on Neural Information Processing Systems (NeurIPS 2021).

models can be a good metric that measures quality of samples [6], and thus can be used for scoring generated samples given by implicit model or detecting outliers as well as noises in input true samples [53]. On the other hand, the generated samples from implicit models could augment the dataset and help to alleviate mode collapse especially when true samples are insufficient that would possibly make explicit model fail to capture an accurate distribution. We refer to Appendix A for a more comprehensive literature review.

Motivated by the discussions above, in this paper, we propose a joint learning framework that enables mutual calibration between explicit and implicit generative models. In our framework, an explicit model is used to estimate the unnormalized density; in the meantime, an implicit generator model is exploited to minimize certain statistical distance (such as the Wasserstein metric or Jensen-Shannon divergence) between the distributions of the true and the generated samples. On top of these two models, a Stein discrepancy, acting as a *bridge* between generated samples and estimated densities, is introduced to push the two models to achieve a consensus. Unlike flow-based models [36, 26], our formulation does not impose invertibility constraints on the generative models and thus is flexible in utilizing general neural network architectures. Our main contributions are as follows:

i) Theoretically, we prove that our method allows the two generative models to impose novel mutual regularization on each other. Specifically, our formulation penalizes large kernel Sobolev norm of the critic in the implicit (WGAN) model, which ensures the critic not to change suddenly on the high-density regions and thus preventing the critic of the implicit model being too strong during training. In the mean time, our formulation also smooths the function given by the Stein discrepancy through Moreau-Yosida regularization, which encourages the explicit model to seek more modes in the data distribution and thus alleviates mode collapse.

ii) In addition, we show that our joint training helps to stabilize the training dynamics. Compared with other common regularization approaches for GAN models that may shift original optimum, our method can facilitate convergence to unbiased model distribution.

iii) We conduct comprehensive experiments to justify our theoretical findings and demonstrate that joint training can help two models achieve better performance. On the one hand, the energy model can detect complicated modes in data more accurately and distinguish out-of-distribution samples. On the other hand, the implicit model can generate higher-quality samples.

## 2 Background

**Energy Model.** The energy model assigns each data $\mathbf{x} \in \mathbb{R}^d$ with a scalar energy value $E_\phi(\mathbf{x})$, where $E_\phi(\cdot)$ is called energy function and parameterized by $\phi$. The model is expected to assign low energy to true samples according to a Gibbs distribution $p_\phi(\mathbf{x}) = \exp\{-E_\phi(\mathbf{x})\}/Z_\phi$, where $Z_\phi$ is a normalizing constant dependent of $\phi$. The term $Z_\phi$ is often hard to compute, making optimization intractable, and various methods are proposed to detour such term (see Appendix A).

**Stein Discrepancy.** Stein discrepancy [16, 30, 5, 38, 3] is a measure of closeness between two probability distributions and does not require knowledge for the normalizing constant of one of the compared distributions. Let $\mathbb{P}$ and $\mathbb{Q}$ be two probability distributions on $\mathcal{X} \subset \mathbb{R}^d$, and assume $\mathbb{Q}$ has a (unnormalized) density $q$. The Stein discrepancy $\mathcal{S}(\mathbb{P}, \mathbb{Q})$ is defined as

$$\mathcal{S}(\mathbb{P}, \mathbb{Q}) := \sup_{\mathbf{f} \in \mathcal{F}} \mathbb{E}_{\mathbf{x} \sim \mathbb{P}}[\mathcal{A}_{\mathbb{Q}} \mathbf{f}(\mathbf{x})] = \sup_{\mathbf{f} \in \mathcal{F}} \Gamma(\mathbb{E}_{\mathbf{x} \sim \mathbb{P}}[\nabla_{\mathbf{x}} \log q(\mathbf{x}) \mathbf{f}(\mathbf{x})^\top + \nabla_{\mathbf{x}} \mathbf{f}(\mathbf{x})]), \qquad (1)$$

where $\mathcal{F}$ is often chosen to be a Stein class (see, e.g., Definition 2.1 in [30]), $\mathbf{f} : \mathbb{R}^d \to \mathbb{R}^{d'}$ is a vector-valued function called *Stein critic* and $\Gamma$ is an operator that transforms a $d \times d'$ matrix into a scalar value. One common choice of $\Gamma$ is trace operator when $d' = d$. One can also use other forms for $\Gamma$, like matrix norm when $d' \neq d$ [30]. If $\mathcal{F}$ is a unit ball in some reproducing kernel Hilbert space (RKHS) with a positive definite kernel $k$, it induces Kernel Stein Discrepancy (KSD) [17]. More details are in Appendix B.

**Wasserstein Metric.** Wasserstein metric is suitable for measuring distances between two distributions with non-overlapping supports [2]. The Wasserstein-1 metric between $\mathbb{P}$ and $\mathbb{Q}$ is

$$\mathcal{W}(\mathbb{P}, \mathbb{Q}) := \min_\gamma \mathbb{E}_{(\mathbf{x}, \mathbf{y}) \sim \gamma}[\|\mathbf{x} - \mathbf{y}\|],$$

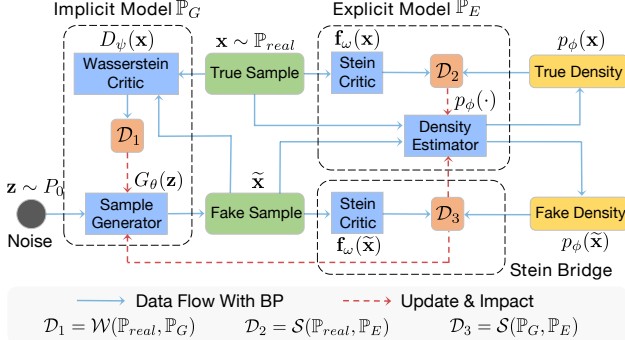

Figure 1: Model framework for *Stein Bridging*.

Table 1: Comparison of objectives between different generative models, where $\mathcal{D}_1 := \mathcal{D}_1(\mathbb{P}_{\mathrm{real}}, \mathbb{P}_G)$, $\mathcal{D}_2 := \mathcal{D}_2(\mathbb{P}_{\mathrm{real}}, \mathbb{P}_E)$ and $\mathcal{D}_3 := \mathcal{D}_3(\mathbb{P}_G, \mathbb{P}_E)$ denote general statistical distances between two distributions.

| Model | Objective |
|---|---|
| GAN [15] | $\mathcal{D}_1$ |
| Energy Model [35] | $\mathcal{D}_2$ |
| Energy-based GAN [55] | $\mathcal{D}_1$ |
| Contrastive Divergence [25] | $\mathcal{D}_2$ |
| Cooperative Learning [50] | $\mathcal{D}_2 + \mathcal{D}_3$ |
| Two Generator Game [7] | $\mathcal{D}_2 + \mathcal{D}_3$ |
| Stein Bridging (ours) | $\mathcal{D}_1 + \mathcal{D}_2 + \mathcal{D}_3$ |

where the minimization with respect to $\gamma$ is over all joint distributions with marginals $\mathbb{P}$ and $\mathbb{Q}$. By Kantorovich-Rubinstein duality, $\mathcal{W}(\mathbb{P}, \mathbb{Q})$ has a dual representation

$$\mathcal{W}(\mathbb{P}, \mathbb{Q}) := \max_D \left\{ \mathbb{E}_{\mathbf{x} \sim \mathbb{P}}[D(\mathbf{x})] - \mathbb{E}_{\mathbf{y} \sim \mathbb{Q}}[D(\mathbf{y})] \right\}, \tag{2}$$

where the maximization is over all 1-Lipschitz continuous functions.

**Sobolev space and Sobolev dual norm.** Let $L^2(\mathbb{P})$ be the Hilbert space on $\mathbb{R}^d$ equipped with an inner product $\langle u, v \rangle_{L^2(\mathbb{P})} := \int_{\mathbb{R}^d} uv \, d\mathbb{P}(\mathbf{x})$. The (weighted) Sobolev space $H^1$ is defined as the closure of $C_0^\infty$, a set of smooth functions on $\mathbb{R}^d$ with compact support, with respect to norm $\|u\|_{H^1} := \left( \int_{\mathbb{R}^d} (u^2 + \|\nabla u\|_2^2) d\mathbb{P}(\mathbf{x}) \right)^{1/2}$, where $\mathbb{P}$ has a density. For $v \in L^2$, its Sobolev dual norm $\|v\|_{H^{-1}}$ is defined by [10]

$$\|v\|_{H^{-1}} := \sup_{u \in H^1} \left\{ \langle v, u \rangle_{L^2} : \int_{\mathbb{R}^d} \|\nabla u\|_2^2 \, d\mathbb{P}(\mathbf{x}) \le 1, \int_{\mathbb{R}^d} u(\mathbf{x}) d\mathbb{P}(\mathbf{x}) = 0 \right\}.$$

The constraint $\int_{\mathbb{R}^d} u(\mathbf{x}) d\mathbf{x} = 0$ is necessary to guarantee the finiteness of the supremum, and the supermum can be equivalently taken over $C_0^\infty$.

## 3 Proposed Model: Stein Bridging

In this section, we formulate our model *Stein Bridging*. A scheme of our framework is illustrated in Figure 1. Denote by $\mathbb{P}_{\mathrm{real}}$ the underlying real distribution from which the data $\{\mathbf{x}\}$ are sampled. The formulation simultaneously learns two generative models – one explicit and one implicit – that represent estimates of $\mathbb{P}_{\mathrm{real}}$. The explicit generative model has a distribution $\mathbb{P}_E$ on $\mathcal{X}$ with explicit probability density proportional to $\exp(-E(\mathbf{x}))$, $\mathbf{x} \in \mathcal{X}$, where $E$ is referred to as an energy function. We focus on energy-based explicit model in model formulation as it does not enforce any constraints or assume specific density forms. For specifications, one can also consider other explicit models, like autoregressive models or directly using some density forms such as Gaussian distribution with given domain knowledge. The implicit model transforms an easy-to-sample random noise $\mathbf{z}$ with distribution $P_0$ via a generator $G$ to a sample $\widetilde{x} = G(\mathbf{z})$ with distribution $\mathbb{P}_G$. Note that for distribution $\mathbb{P}_E$, we have its explicit density without normalizing term, while for $\mathbb{P}_G$ and $\mathbb{P}_{\mathrm{real}}$, we have samples from two distributions. Hence, we can use the Stein discrepancy (that does *not* require the normalizing constant) as a measure of closeness between the explicit distribution $\mathbb{P}_E$ and the real distribution $\mathbb{P}_{\mathrm{real}}$, and use the Wasserstein metric (that only requires only samples from two distributions) as a measure of closeness between the implicit distribution $\mathbb{P}_G$ and the real data distribution $\mathbb{P}_{\mathrm{real}}$.

To jointly learn the two generative models $\mathbb{P}_G$ and $\mathbb{P}_E$, arguably a straightforward way is to minimize the sum of the Stein discrepancy and the Wasserstein metric:

$$\min_{E, G} \mathcal{W}(\mathbb{P}_{\mathrm{real}}, \mathbb{P}_G) + \lambda \mathcal{S}(\mathbb{P}_{\mathrm{real}}, \mathbb{P}_E),$$

where $\lambda \ge 0$. However, this approach appears no different than learning the two generative models separately. To achieve information sharing between two models, we incorporate another term

$\mathcal{S}(\mathbb{P}_G, \mathbb{P}_E)$ – called the *Stein bridge* – that measures the closeness between the explicit distribution $\mathbb{P}_E$ and the implicit distribution $\mathbb{P}_G$:

$$\min_{E,G} \ \mathcal{W}(\mathbb{P}_{\text{real}}, \mathbb{P}_G) + \lambda_1 \mathcal{S}(\mathbb{P}_{\text{real}}, \mathbb{P}_E) + \lambda_2 \mathcal{S}(\mathbb{P}_G, \mathbb{P}_E), \tag{3}$$

where $\lambda_1, \lambda_2 \geq 0$. The Stein bridge term in (3) pushes the two models to achieve a consensus.

**Remark 1**. Our formulation is flexible in choosing both the implicit and explicit models. In (3), we can choose statistical distances other than the Wasserstein metric $\mathcal{W}(\mathbb{P}_{\text{real}}, \mathbb{P}_G)$ to measure closeness between $\mathbb{P}_{\text{real}}$ and $\mathbb{P}_G$, such as Jensen-Shannon divergence, as long as its computation requires only samples from the involved two distributions. Hence, one can use GAN architectures other than WGAN to parametrize the implicit model. In addition, one can replace the first Stein discrepancy term $\mathcal{S}(\mathbb{P}_{\text{real}}, \mathbb{P}_E)$ in (3) by other statistical distances as long as its computation is efficient and hence other explicit models can be used. For instance, if the normalizing constant of $\mathbb{P}_E$ is known or easy to calculate, one can use Kullback-Leibler (KL) divergence.

**Remark 2**. The choice of the Stein discrepancy for the bridging term $\mathcal{S}(\mathbb{P}_G, \mathbb{P}_E)$ is crucial and cannot be replaced by other statistical distances such as KL divergence, since the data-generating distribution does not have an explicit density form (not even up to a normalizing constant). This is exactly one important reason why Stein bridging was proposed, which requires only samples from the data distribution and only the log-density of the explicit model without the knowledge of normalizing constant as estimated in MCMC or other methods.

In our implementation, we parametrize the generator in implicit model and the density estimator in explicit model as $G_\theta(\mathbf{z})$ and $p_\phi(\mathbf{x})$, respectively. The Wasserstein term in (3) is implemented using its equivalent dual representation in (2) with a parametrized critic $D_\psi(\mathbf{x})$. The two Stein terms in (3) can be implemented using (1) with either a Stein critic (parametrized as a neural network, i.e., $\mathbf{f}_w(\mathbf{x})$), or the non-parametric Kernel Stein Discrepancy. Our implementation iteratively updates the explicit and implicit models. Details for model specifications and optimization are in Appendix E.2.

**Comparison with Existing Works.** There are several studies that attempt to combine explicit and implicit generative models from different ways, e.g. by energy-based GAN [55], contrastive divergence [25, 6], cooperative learning [50] or two generator game [7]. Here we provide a high-level comparison in Table 1 where we note that the formulations of existing works only consider one-side discrepancy or at most two discrepancy terms. Such formulations cannot address the respective issues for both models and, even worse, the training for two models would constrain rather than exactly compensate each other (more discussions are in Appendix A.3). Differently, our model considers three discrepancies simultaneously as a triangle to jointly optimize two generative models. In the following, we will show that such new simple formulation enables two models to compensate each other via mutual regularization effects and stabilize the training dynamics.

## 4 Theoretical Analysis

In this section, we provide theoretical insights on proposed scheme, which illuminate its mutual regularization effects as a justification of our joint training and further show its merit for stabilizing the training dynamics. The proofs for all the results in this section are in Appendix D.

### 4.1 Mutual Regularization Effects.

We first show the regularization effect of the Stein bridge on the Wasserstein critic. Define the *kernel Sobolev dual norm* as

$$\|D\|_{H^{-1}(\mathbb{P};k)} := \sup_{u \in C_0^\infty} \{ \langle D, u \rangle_{L^2(\mathbb{P})} : \mathbb{E}_{\mathbf{x},\mathbf{x}' \sim \mathbb{P}} [\nabla u(\mathbf{x})^\top k(\mathbf{x}, \mathbf{x}') \nabla u(\mathbf{x}')] \leq 1, \ \mathbb{E}_{\mathbb{P}}[u] = 0 \}.$$

It can be viewed as a kernel generalization of the Sobolev dual norm defined in Section 2, which reduces to the Sobolev dual norm when $k(\mathbf{x}, \mathbf{x}') = \mathbb{I}(\mathbf{x} = \mathbf{x}')$ and $\mathbb{P}$ is the Lebesgue measure.

**Theorem 1.** *Assume that $\{\mathbb{P}_G\}_G$ exhausts all continuous probability distributions and $\mathcal{S}$ is chosen as kernel Stein discrepancy. Then problem (3) is equivalent to*

$$\min_E \max_D \left\{ \mathbb{E}_{\mathbf{y} \sim \mathbb{P}_E}[D(\mathbf{y})] - \mathbb{E}_{\mathbf{x} \sim \mathbb{P}_{\text{real}}}[D(\mathbf{x})] - \frac{1}{4\lambda_2} \|D\|_{H^{-1}(\mathbb{P}_E;k)}^2 + \lambda_1 \mathcal{S}(\mathbb{P}_{\text{real}}, \mathbb{P}_E) \right\}.$$

The kernel Sobolev norm regularization penalizes large variation of the Wasserstein critic $D$. Particularly, observe that [45] if $k(\mathbf{x}, \mathbf{x}') = \mathbb{I}(\mathbf{x} = \mathbf{x}')$ and $\mathbb{E}_{\mathbb{P}_E}[D] = 0$, and then

$$\|D\|_{H^{-1}(\mathbb{P}_E; k)} = \lim_{\epsilon \to 0} \frac{\mathcal{W}_2((1 + \epsilon D)\mathbb{P}_E, \mathbb{P}_E)}{\epsilon},$$

where $\mathcal{W}_2$ denotes the 2-Wasserstein metric. Hence, the Sobolev dual norm regularization ensures $D$ not to change suddenly on high-density region of $\mathbb{P}_E$, and thus reinforces the learning of the Wasserstein critic. Stein bridge penalizes large variation of the Wasserstein critic, in the same spirit but of different form comparing to gradient-based penalty (e.g., [19, 40]). It prevents Wasserstein critic from being too strong during training and thus encourages mode exploration of sample generator. To illustrate this, we conduct a case study where we train a generator over the data sampled from a mixture of Gaussian ($\mu_1 = [-1, -1]$, $\mu_2 = [1, 1]$ and $\Sigma = 0.2\mathbf{I}$). In Fig. 3(a) we compare gradient norms of the Wasserstein critic when training the generator with and without the Stein bridge. As we can see, Stein bridge can help to reduce gradient norms, with a similar effect as WGAN-GP.

Moreover, the Stein bridge also plays a part in smoothing the output from Stein discrepancy and we show the result in the following theorem.

**Theorem 2.** *Assume $\{\mathbb{P}_G\}_G$ exhausts all continuous probability distributions, and the Stein class defining the Stein discrepancy is compact (in some linear topological space). Then problem (3) is equivalent to*

$$\min_E \left\{ \lambda_1 \mathcal{S}(\mathbb{P}_{\text{real}}, \mathbb{P}_E) + \lambda_2 \max_{\mathbf{f}} \mathbb{E}_{\mathbf{x} \sim \mathbb{P}_{\text{real}}}[(\mathcal{A}_{\mathbb{P}_E}\mathbf{f})_{\lambda_2}(\mathbf{x})] \right\},$$

*where $(\mathcal{A}_{\mathbb{P}_E}\mathbf{f})_{\lambda_2}(\cdot)$ denotes the (generalized) Moreau-Yosida regularization of function $\mathcal{A}_{\mathbb{P}_E}\mathbf{f}$ with parameter $\lambda_2$, i.e., $(\mathcal{A}_{\mathbb{P}_E}\mathbf{f})_{\lambda_2}(\mathbf{x}) = \min_{\mathbf{y} \in \mathcal{X}}\{\mathcal{A}_{\mathbb{P}_E}\mathbf{f}(\mathbf{y}) + \frac{1}{\lambda_2}\|\mathbf{x} - \mathbf{y}\|\}$.*

Note that $(\mathcal{A}_{\mathbb{P}_E}\mathbf{f})_{\lambda_2}$ is Lipschitz continuous with constant $1/\lambda_2$. Hence, the Stein bridge, together with the Wasserstein metric $\mathcal{W}(\mathbb{P}_{\text{real}}, \mathbb{P}_G)$, plays as a Lipschitz regularization on the output of the Stein operator $\mathcal{A}_{\mathbb{P}_E}\mathbf{f}$ via Moreau-Yosida regularization. This suggests a novel regularization scheme for Stein-based GAN. By smoothing the Stein critic, the Stein bridge encourages the energy model to seek more modes in data instead of focusing on some dominated modes, thus alleviating mode-collapse issue.

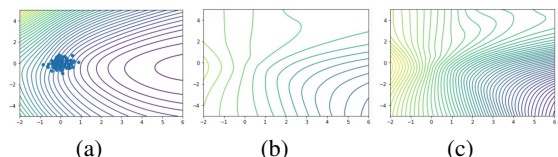

(a)  (b)  (c)

Figure 2: (a) Contour of an energy model with one mode and empirical data from a distribution with a different mode (blue dots); (b) & (c) Contours of the Stein critics between the two distributions learned w/ and w/o the Stein bridge, respectively.

To illustrate this, we consider a case where we have an energy model initialized with one mode center and data sampled from distribution of another mode, as depicted in Fig. 2(a). Fig. 2(b) and 2(c) compare the Stein critics when using Stein bridge and not, respectively. The Stein bridge helps to smooth the Stein critic, as indicated by a less rapidly changing contour in Fig. 2(b) compared to Fig. 2(c), learned from the data and model distributions plotted in Fig. 2(b).

## 4.2 Stability of training dynamics.

We further show that Stein Bridging could stabilize adversarial training between generator and Wasserstein critic with a local convergence guarantee. As is known, the training for minimax game in GAN is difficult. When using traditional gradient methods, the training would suffer from some oscillatory behaviors [14, 29, 54]. In order to better understand the optimization behaviors, we first compare the behaviors of WGAN, likelihood- and entropy-regularized WGAN, and our Stein Bridging under SGD via an easy to comprehend toy example in one-dimensional case. Such a toy example (or a similar one) is also utilized by [13, 33] to shed lights on the instability of WGAN training[2]. Consider a linear critic $D_\psi(x) = \psi x$ and generator $G_\theta(z) = \theta x$. Then the Wasserstein GAN objective can be written as a constrained bilinear problem: $\min_\theta \max_{|\psi| \leq 1} \psi \mathbb{E}[x] - \psi\theta\mathbb{E}[z]$, which could be further simplified as an unconstrained version (the behaviors can be generalized to multi-dimensional cases [13]):

$$\min_\theta \max_\psi \psi - \psi \cdot \theta. \tag{4}$$

---

[2]Our theoretical discussions focus on WGAN, and we also compare with original GAN in the experiments.

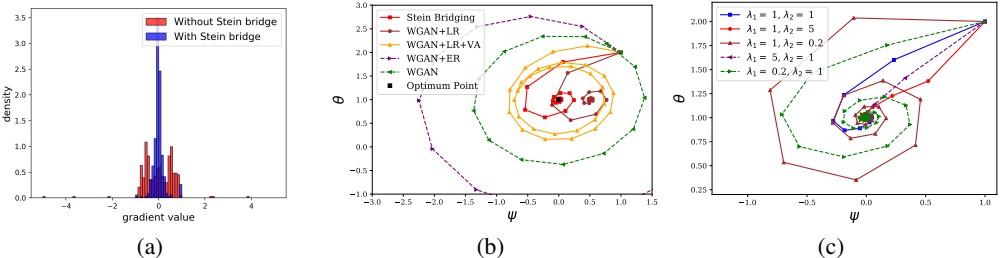

(a)                    (b)                    (c)

Figure 3: (a) The gradient norm of Wasserstein critic with (blue) and without (red) the Stein bridge when data are sampled from a mixture of Gaussian. (b) Numerical SGD updates of Stein Bridging, WGAN and its variants with different regularizations.

Unfortunately, such simple objective cannot guarantee convergence by traditional gradient methods like SGD with alternate updating[3]: $\theta_{k+1} = \theta_k + \eta\psi_k, \psi_{k+1} = \psi_k + \eta(1-\theta_{k+1})$. Such optimization would suffer from an oscillatory behavior, i.e., the updated parameters go around the optimum point ($[\psi^*, \theta^*] = [0,1]$) forming a circle without converging to the centrality, which is shown in Fig. 3(b). A recent study in [29] theoretically show that such oscillation is due to the interaction term in (4).

One solution to the instability of GAN training is to add (likelihood) regularization, which has been widely studied by recent literatures [46, 28]. With regularization term, the objective changes into $\min_\theta \max_{|\psi|\leq 1} \psi\mathbb{E}[x] - \psi\theta\mathbb{E}[z] - \lambda\mathbb{E}[\log\mu(\theta z)]$, where $\mu(\cdot)$ denotes the likelihood function and $\lambda$ is a hyperparameter. A recent study [44] proves that when $\lambda < 0$ (likelihood-regularization), the extra term is equivalent to maximizing sample evidence, helping to stabilize GAN training; when $\lambda > 0$ (entropy-regularization), the extra term maximizes sample entropy, which encourages diversity of generator. Here we consider a Gaussian likelihood function for generated sample $x'$, $\mu(x') = \exp(-\frac{1}{2}(x'-b)^2)$ which is up to a constant, and the objective becomes (see Appendix D.1 for details):

$$\min_\theta \max_\psi \psi - \psi \cdot \theta - \lambda(\theta^2 - \theta). \tag{5}$$

The above system would converge with $\lambda < 0$ and diverge with $\lambda > 0$ in gradient-based optimization, shown in Fig. 3(b). Another issue of likelihood-regularization is that the extra term changes the optimum point and makes the model converge to a biased distribution, as proved by [44]. In this case, one can verify that the optimum point becomes $[\psi^*, \theta^*] = [-\lambda, 1]$, resulting in a bias. To avoid this issue, [44] proposes to temporally decrease $|\lambda|$ through training. However, such method would also be stuck in oscillation when $|\lambda|$ gets close to zero as is shown in Fig. 3(b).

Finally, consider our proposed model. We also simplify the density estimator as a basic energy model $p_\phi(x) = \exp(-\frac{1}{2}x^2 - \phi x)$ whose score function $\nabla_x \log p_\phi(x) = -x - \phi$. Then if we specify the two Stein discrepancies in (3) as KSD, the objective is (see Appendix D.1 for details),

$$\min_\theta \max_\psi \min_\phi \psi - \psi \cdot \theta + \frac{\lambda_1}{2}(1+\phi)^2 + \frac{\lambda_2}{2}(\theta+\phi)^2. \tag{6}$$

Interestingly, for $\forall\lambda_1, \lambda_2$, the optimum remains the same $[\psi^*, \theta^*, \phi^*] = [0, 1, -1]$. Then we show that the optimization guarantees convergence to $[\psi^*, \theta^*, \phi^*]$.

**Proposition 1.** *Using alternate SGD for (6) geometrically decreases the square norm $N_t = |\psi^t|^2 + |\theta-1|^2 + |\phi+1|^2$, for any $0 < \eta < 1$ with $\lambda_1 = \lambda_2 = 1$,*

$$N_{t+1} = (1 - \eta^2(1-\eta)^2)N_t. \tag{7}$$

As shown in Fig. 3(b), Stein Bridging achieves a good convergence to the right optimum. Compared with (4), the objective (6) adds a new bilinear term $\phi \cdot \theta$, which acts like a connection between the generator and estimator, and two other quadratic terms, which help to penalize the increasing of values through training. The added terms and original terms in (6) cooperate to guarantee convergence to a unique optimum. (More discussions in Appendix D.1). Moreover, Fig. 3(c) presents the training dynamics w.r.t. different $\lambda_1$'s and $\lambda_2$'s. As we can see, the convergence can be achieved with different trading-off parameters which in essence have impact on the convergence speed.

---

[3]Here, we adopt the most widely used alternate updating strategy. The simultaneous updating, i.e., $\theta_{k+1} = \theta_k + \eta\psi_k$ and $\psi_{k+1} = \psi_k + \eta(1-\theta_k)$, would diverge in this case.

We further generalize the analysis to multi-dimensional bilinear system $F(\boldsymbol{\psi}, \boldsymbol{\theta}) = \boldsymbol{\theta}^\top \mathbf{A} \boldsymbol{\psi} - \mathbf{b}^\top \boldsymbol{\theta} - \mathbf{c}^\top \boldsymbol{\psi}$ which is extensively used by researches for analysis of GAN stability [14, 12, 29, 13]. For any bilinear system, with added term $H(\boldsymbol{\phi}, \boldsymbol{\theta}) = \frac{1}{2}(\boldsymbol{\theta} + \boldsymbol{\phi})^\top \mathbf{B}(\boldsymbol{\theta} + \boldsymbol{\phi})$ where $\mathbf{B} = (\mathbf{A}\mathbf{A}^\top)^{\frac{1}{2}}$ to the objective, we can prove that i) the optimum point remains the same as the original system (Proposition 3) and ii) using alternate SGD algorithm for the new objective can guarantee convergence (Theorem 3). More discussions are given in Appendix D.2.

## 5   Experiments

In this section, we conduct experiments to verify the effectiveness of proposed method from multifaceted views. The implementation codes are available at `https://github.com/qitianwu/SteinBridging`.

### 5.1   Setup

We mainly consider evaluation with two tasks: density estimation and sample generation. For density estimation, we expect the model to output estimated density values for input samples and the estimation is supposed to match the ground-truth one. For sample generation, the model aims at generating samples that are akin to the real observed ones.

We consider two synthetic datasets with mixtures of Gaussian distributions: Two-Circle and Two-Spiral. The first one is composed of 24 Gaussian mixtures that lie in two circles. The second dataset consists of 100 Gaussian mixtures densely arranged on two centrally symmetrical spiral-shaped curves. The ground-truth distributions are shown in Fig. 4(a). Details for synthetic datasets are in Appendix E.1. Furthermore, we apply the method to MNIST and CIFAR datasets which require the model to deal with high-dimensional image data.

In each dataset, we use true observed samples as input of the model and leverage them to train our model. In synthetic datasets, we sample $N_1 = 2000$ and $N_2 = 5000$ points from the ground-truth distributions as true samples for Two-Circle and Two-Spiral datasets, respectively. The true samples are shown in Fig. 4 (a). In MNIST and CIFAR, we directly use pictures in the training sets as true samples. The details for each dataset are reported in Appendix E.1.

We term our model *Joint-W* if using Wasserstein metric in (3) and *Joint-JS* if using JS divergence in this section. As we mentioned, our model is capable for 1) yielding estimated (unnormalized) density values (by the explicit energy model) for input samples and 2) generating samples (by the implicit generative model) from a noise distribution. We consider several competitors for performance comparison. For sample generation, we mainly compare our model with implicit generative models. Specifically, we basically consider the counterparts without joint training with energy model, which are equivalently valina GAN and WGAN with gradient penalty [19], for ablation study. Also, as comparison to the new regularization effects by Stein Bridging, we consider a recently proposed variational annealing regularization [44] for GANs (short as GAN+VA/WGAN+VA) with denoising auto-encoder [1] to estimate the gradient for regularization penalty. For density estimation, we mainly compare with explicit models. Specifically, we also consider the counterparts without joint training with generator model, i.e., Deep Energy Model (DEM) using Stein discrepancy [18]. Besides we compare with energy calibrated GAN (EGAN) [6] and Deep Directed Generative (DGM) Model [25] which adopt contrastive divergence to train a sample generator with an energy estimator. The hyperparameters are tuned according to quantitative metrics (will be discussed later) used for different tasks. See Appendix E.3 for implementation details.

### 5.2   Density Estimation of Explicit Model

As shown in Two-Circle case in Fig 5, both Joint-JS and Joint-W manage to capture all Gaussian components while other methods miss some of modes. In Two-Spiral case in Fig 4, Joint-JS and Joint-W exactly fit the ground-truth distribution. Nevertheless, DEM misses one spiral while EGAN degrades to a uniform-like distribution. DGM manages to fit two spirals but allocate high densities to regions that have low densities in the groung-truth distribution. As quantitative comparison, we study three evaluation metrics: KL & JS divergence and Area Under the Curve (AUC). Detailed information and results are in Appendix E.4 and Table 6 respectively. The values show that Joint-W and Joint-JS provide better density estimation than all the competitors over a large margin.

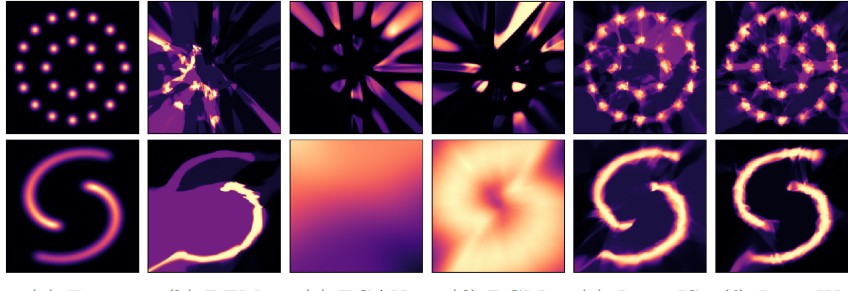

|       |       |       |       |       |       |
| (a) True | (b) DEM | (c) EGAN | (d) DGM | (e) Joint-JS | (f) Joint-W |

Figure 4: Results for density estimation. (a) Densities of real distribution. (b)∼(f) Estimated densities given by the estimators of different methods on Two-Circle (upper) and Two-Spiral (bottom) datasets.

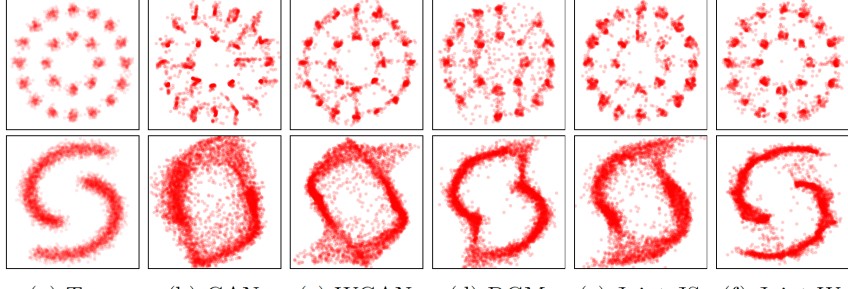

|       |       |       |       |       |       |
| (a) True | (b) GAN | (c) WGAN | (d) DGM | (e) Joint-JS | (f) Joint-W |

Figure 5: Comparison for sample quality. (a) Samples from real distribution. (b)∼(f) Generated samples produced by the generators of different methods on Two-Circle (upper) and Two-Spiral (bottom) datasets.

We rank generated digits (and true digits) on MNIST w.r.t densities given by the energy model in Fig. 13, Fig. 14 and Fig. 15. As depicted in the figures, the digits with high densities (or low densities) given by Joint-JS possess enough diversity (the thickness, the inclination angles as well as the shapes of digits diverses). By contrast, all the digits with high densities given by DGM tend to be thin and digits with low densities are very thick. Also, as for EGAN, digits with high (or low) densities appear to have the same inclination angle (for high densities, '1' keeps straight and '9' 'leans' to the left while for low densities, just the opposite), which indicates that DGM and EGAN tend to allocate high (or low) densities to data with certain modes and miss some modes that possess high densities in ground-truth distributions. By contrast, our method manages to capture these complicated features in data distributions.

We further study model performance on detection for out-of-distribution (OOD) samples. We consider CIFAR-10 images as positive samples and construct negative samples by (I) flip images, (II) add random noise, (III) overlay two images and (IV) use images from LSUN dataset, respectively. A good density models trained on CIFAR-10 are expected to give high densities to positive samples and low densities to negative samples, with exception for case (I) (flipping images are not exactly negative samples and the model should give high densities). We use the density values rank samples and calculate AUC of false positive rate v.s. true positive rate, reported in Table 3. Our model Joint-W manages to distinguish samples for (II), (III), (IV) and is not fooled by flipping images, while DEM and EGAN fail to detect out-of-distribution samples and DGM gives wrong results, recognizing flipping images as negative samples.

### 5.3 Sample Quality of Implicit Model

In Fig. 5 we show the results of different generators in synthetic datasets. For Two-Circle, there are a large number of generated samples given by GAN, WGAN-GP and DGM locating between two Gaussian components, and the boundary for each component is not distinguishable. Since the ground-truth densities of regions between two components are very low, such generated samples possess low-quality, which depicts that these models capture the combinations of two dominated features (i.e., modes) in data but such combination makes no sense in practice. By contrast, Joint-JS and Joint-W could alleviate such issue, reduce the low-quality samples and produce more distinguishable

Table 2: Inception Scores (IS) and Fréchet Inception Distance (FID) on CIFAR-10 datasets.

| Method | IS | FID |
|---|---|---|
| WGAN-GP | 6.74±0.041 | 42.2±0.572 |
| Energy GAN | 6.89±0.081 | 45.6±0.375 |
| WGAN+VA | 6.90±0.058 | 45.3±0.307 |
| DGM | 6.51±0.041 | 48.8±0.492 |
| **Joint-W(ours)** | **7.12**±0.101 | **41.0**±0.546 |

Table 3: Area Under the Curve (AUC) for OOD detection in CIFAR-10 datasets.

| Method | I | II | III | IV |
|---|---|---|---|---|
| DEM | 0.50 | 0.52 | 0.51 | 0.56 |
| DGM | 1.00 | 1.00 | 1.00 | 0.82 |
| EGAN | 0.50 | 0.42 | 0.30 | 0.52 |
| **Joint-W** | 0.50 | 0.92 | 0.95 | 0.85 |

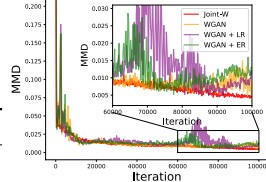

Figure 6: Learning curves in Two-Spiral.

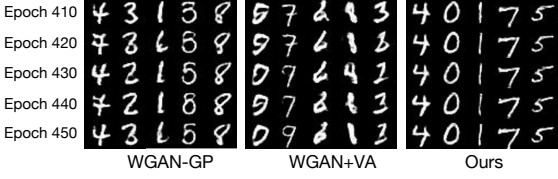

Figure 7: Generated digits given by the same noise $z$ in adjacent training epochs on MNIST.

Figure 8: Impact of (a) noise in data and (b) insufficient data on model performance.

boundaries. In Two-Spiral, similarly, the generated samples given by GAN and WGAN-GP form a circle instead of two spirals while the samples of DGM 'link' two spirals. Joint-JS manages to focus more on true high densities compared to GAN and Joint-W provides the best results. To quantitatively measure sample quality, we adopt Maximum Mean Discrepancy (MMD) and High-quality Sample Rate (HSR). Details are in Appendix E.4 and results are in Table 6 where our models can outperform the competitors over a large margin.

We report the Inception Score (IS) and Fréchet Inception Distance (FID) to measure the sample quality on CIFAR-10. As shown in Table 2, Joint-W outperforms other competitors by 0.2 and achieves 5.6% improvement over WGAN-GP w.r.t IS. As for FID, Joint-W slightly outperforms WGAN-GP and beats energy-based GAN and variational annealing regularized WGAN over a large margin. One possible reason is that these methods both consider entropy regularization which encourages diversity of generated samples but will have a negative effect on sample quality. Stein Bridging can overcome this issue via joint training with explicit model. In practice, DGM is hard for convergence during training and gives much worse performance than others.

### 5.4 Further Discussions

**Enhancing the Stability of GAN.** In Fig. 6 we present the learning curve of Joint-W compared with WGAN and likelihood- and entropy-regularized WGAN. The curves depict that joint training could reduce the variance of metric values especially during the second half of training. Furthermore, we visualize generated digits given by the same noise $z$ in adjacent epochs in Fig. 7. The results show that Joint-W gives more stable generation in adjacent epochs while generated samples given by WGAN-GP and WGAN+VA exhibit an obvious variation. Especially, some digits generated by WGAN-GP and WGAN+VA change from one class to another, which is quite similar to the oscillation without convergence discussed in Section 3.2. To quantify the evaluation of bias in model distributions, we calculate distances between the means of 50000 generated digits (resp. images) and 50000 true digits (resp. images) in MNIST (reps. CIFAR-10). The results are reported in Table 5. We can see that the model distributions of other methods are more seriously biased from true distribution, compared with Joint-W.

**Contaminated or Limited Data.** In Fig. 8(a) we compare Joint-W with WGAN-GP for sample generation on noisy input in Two-Circle dataset. Details are in Appendix E.1. We can see that the noise ratio in data impacts the performance of WGAN-GP and Joint-W, but comparatively, the performance decline of Joint-W is less significant, which indicates better robustness of joint training w.r.t. noised data. Moreover, in Fig. 8(b), we compare Joint-W with DEM for density estimation with insufficient true data in Two-Spiral dataset. When sample size decreases from 2000 to 100, the AUC value of DEM declines dramatically. By contrast, the AUC of Joint-W exhibits a small decline when the sample size is more than 500. The results demonstrate that Stein Bridging has promising power in some extreme cases where the training sample are contaminated or limited.

# 6 Conclusions and Discussions

This paper aims at jointly training implicit generative model and explicit generative model via an bridging term of Stein discrepancy. Theoretically, we show that joint training could i) enforce dual regularization effects on both models and thus encourage mode exploration, and ii) help to facilitate the convergence of minimax training dynamics. Extensive experiments on various tasks show that our method can achieve better performance on both sample generation and density estimation.

**Limitations.** We mainly focus on GAN/WGAN as instantiations of implicit generative models and energy-based models as instantiations of explicit models for theoretical analysis and empirical evaluation. In fact, our formulation can also be extended to other models like VAE, flow model, etc. to combine the best of two worlds. Furthermore, since we propose a new learning paradigm as a piorneering endeavor on unifying the training of two generative models via concurrently optimizing three loss terms, our experiments mainly focus on synthetic datasets, MNIST and CIFAR-10. We believe our method can be applied to more complicated high-dimensional datasets given the promising results in this paper.

**Potential Societal Impacts.** One merit of our method is that the joint training model makes it easier to add inductive bias to the generative models, as discussed in Section 1 and 4.4. Such inductive bias enforced manually can be used to control the distribution of output samples with some desirable properties that accord with ethical considerations, e.g., fairness. Admittedly, such inductive bias would also possibly be used by some speculators for generating false works for commercial purposes. More studies are needed in the future to detect such works generated by machines in a more intelligent way and further protect intellectual property of individuals.

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
