# A  Literature Reviews

We discuss some of related literature and shed lights on the relationship between our work with others.

## A.1  Explicit Generative Models

Explicit generative models are interested in fitting each instance with a scalar (unnormalized) density expected to explicitly capture the distribution behind data. Such densities are often up to a constant and called as energy functions which are common in undirected graphical models [24]. Hence, explicit generative models are also termed as energy-based models. An early version of energy-based models is the FRAME (Filters, Random field, And Maximum Entropy) model [53, 45]. Later on, some works leverage deep neural networks to model the energy function [32, 49] and pave the way for researches on deep energy model (DEM) (e.g., [29, 22, 50, 17, 8, 34]). Apart from DEM, there are also some other forms of deep explicit models based on deep belief networks [19] and deep Boltzmann machines [38].

The normalized constant under the energy function requires an intractable integral over all possible instances, which makes the model hard to learn via Maximum Likelihood Estimation (MLE). To solve this issue, some works propose to approximate the constant by MCMC methods [10, 31]. However, MCMC requires an inner-loop samples in each training, which induces high computational costs. Another solution is to optimize an alternate surrogate loss function. For example, contrastive divergence (CD) [29] is proposed to measure how much KL divergence can be improved by running a small numbers of Markov chain steps towards the intractable likelihood, while score matching (SM) [21] detours the constant by minimizing the distance for gradients of log-likelihoods. A recent study [15] uses Stein discrepancy to train unnormalized model. The Stein discrepancy does not require the normalizing constant and makes the training tractable. Moreover, the intractable normalized constant makes it hard to sample from. To obtain an accurate samples from unnormalized densities, many studies propose to approximate the generation by diffusion-based processes, like generative flow [33] and variational gradient descent ([28]). Also, a recent work [20] leverages Stein discrepancy to design a neural sampler from unnormalized densities. The fundamental disadvantage of explicit model is that the energy-based learning is difficult to accurately capture the distribution of true samples due to the low manifold of real-world instances [29].

## A.2  Implicit Generative Models

Implicit generative models focus on a generation mapping from random noises to generated samples. Such mapping function is often called as generator and possesses better flexibility compared with explicit models. One typical implicit model is Generative Adversarial Networks (GAN) [14]. GAN targets an adversarial game between the generator and a discriminator (or critic in WGAN) that aims at discriminating the generated and true samples. In this paper, we focus on GAN and its variants (e.g., WGAN [2], WGAN-GP [16], DCGAN [36], etc.) as the implicit generative model and we leave the discussions on other implicit models as future work.

Two important issues concerning GAN and its variants are instability of training and local optima. The typical local optima for GAN can be divided into two categories: mode-collapse (the model fails to capture all the modes in data) and mode-redundance (the model generates modes that do not exist in data). Recently there are many attempts to solve these issues from various perspectives. One perspective is from regularization. Two typical regularization methods are likelihood-based and entropy-based regularization with the prominent examples [43] and [25] that respectively leverage denoising feature matching and implicit gradient approximation to enforce the regularization constraints. The likelihood and entropy regularizations could respectively help the generator to focus on data distribution and encourage more diverse samples, and a recent work [41] uses Langevin dynamics to indicate that i) the entropy and likelihood regularizations are equivalent and share an opposite relationship in mathematics, and ii) both regularizations would make the model converge to a surrogate point with a bias from original data distribution. Then [41] proposes a variational annealing strategy to empirically unite two regularizations and tackle the biased distributions.

To deal with the instability issue, there are also some recent literatures from optimization perspectives and proposes different algorithms to address the non-convergence of minimax game optimization (for

instance, [11, 26, 12]). Moreover, the disadvantage of implicit models is the lack of explicit densities over instances, which disables the black-box generator to characterize the distributions behind data.

## A.3 Attempts to Combine Both of the Worlds

Recently, there are several studies that attempt to combine explicit and implicit generative models from different ways. For instance, [52] proposes energy-based GAN that leverages energy model as discriminator to distinguish the generated and true samples. The similar idea is also used by [22] and [5] which let the discriminator estimate a scaler energy value for each sample. Such discriminator is optimized to give high energy to generated samples and low energy to true samples while the generator aims at generating samples with low energy. The fundamental difference is that [52] and [5] both aim at minimizing the discrepancy between distributions of generated and true samples while the motivation of [22] is to minimize the KL divergence between estimated densities and true samples. [22] adopts contrastive divergence (CD) to link MLE for energy model over true data with the adversarial training of energy-based GAN. However, both CD-based method and energy-based GAN have limited power for both generator and discriminator. Firstly, if the generated samples resemble true samples, then the gradients for discriminator given by true and generated samples are just the opposite and will counteract each other, and the training will stop before the discriminitor captures accurate data distribution. Second, since the objective boils down to minimizing the KL divergence (for [22]) or Wasserstein distance (for [5]) between model and true distributions, the issues concerning GAN (or WGAN) like training instability and mode-collapse would also bother these methods.

Another way for combination is by cooperative training. [47] (and its improved version [46]) leverages the samples of generator as the MCMC initialization for energy-based model. The synthesized samples produced from finite-step MCMC are closer to the energy model and the generator is optimized to make the finite-step MCMC revise its initial samples. Also, a recent work [7] proposes to regard the explicit model as a teacher net who guides the training of implicit generator as a student net to produce samples that could overcome the mode-collapse issue. The main drawback of cooperative training is that they indirectly optimize the discrepancy between the generator and data distribution via the energy model as a 'mediator', which leads to a fact that once the energy model gets stuck in a local optimum (e.g., mode-collapse or mode-redundance) the training for the generator would be affected. In other words, the training for two models would constrain rather than exactly compensate each other. Different from existing methods, our model considers three discrepancies simultaneously as a triangle to jointly train the generator and the estimator, enabling them to compensate and reinforce each other.

# B  Background for Stein Discrepancy

Assume $q(\mathbf{x})$ to be a continuously differentiable density supported on $\mathcal{X} \subset \mathbb{R}^d$ and $\mathbf{f} : \mathbb{R}^d \to \mathbb{R}^{d'}$ a smooth vector function. Define $\mathcal{A}_q[\mathbf{f}(\mathbf{x})] = \nabla_{\mathbf{x}} \log q(\mathbf{x})\mathbf{f}(\mathbf{x})^\top + \nabla_{\mathbf{x}}\mathbf{f}(\mathbf{x})$ as a Stein operator. If $\mathbf{f}$ is a Stein class (satisfying some mild boundary conditions) then we have the following Stein identity property:

$$\mathbb{E}_{\mathbf{x} \sim q}[\mathcal{A}_q[\mathbf{f}(\mathbf{x})]] = \mathbb{E}_{\mathbf{x} \sim q}[\nabla_{\mathbf{x}} \log q(\mathbf{x})\mathbf{f}(\mathbf{x})^\top + \nabla_{\mathbf{x}}\mathbf{f}(\mathbf{x})] = 0.$$

Such property induces Stein discrepancy between distributions $\mathbb{P} : p(\mathbf{x})$ and $\mathbb{Q} : q(\mathbf{x})$, $\mathbf{x} \in \mathcal{X}$:

$$\mathcal{S}(\mathbb{Q}, \mathbb{P}) = \sup_{\mathbf{f} \in \mathcal{F}} \Gamma(\mathbb{E}_{\mathbf{x} \sim q}[\mathcal{A}_p[\mathbf{f}(\mathbf{x})]]) = \sup_{\mathbf{f} \in \mathcal{F}}\{\Gamma(\mathbb{E}_{\mathbf{x} \sim q}[\nabla_{\mathbf{x}} \log p(\mathbf{x})\mathbf{f}(\mathbf{x})^\top + \nabla_{\mathbf{x}}\mathbf{f}(\mathbf{x})])\}, \quad (8)$$

where $\mathbf{f}$ is what we call *Stein critic* that exploits over function space $\mathcal{F}$ and if $\mathcal{F}$ is large enough then $\mathcal{S}(\mathbb{Q}, \mathbb{P}) = 0$ if and only if $\mathbb{Q} = \mathbb{P}$. Note that in (1), we do not need the normalized constant for $p(\mathbf{x})$ which enables Stein discrepancy to deal with unnormalized density.

If $\mathcal{F}$ is a unit ball in a Reproducing Kernel Hilbert Space (RKHS) with a positive definite kernel function $k(\cdot, \cdot)$, then the supremum in (1) would have a close form (see [27, 4, 35] for more details):

$$\mathcal{S}_K(\mathbb{Q}, \mathbb{P}) = \mathbb{E}_{\mathbf{x}, \mathbf{x}' \sim q}[u_p(\mathbf{x}, \mathbf{x}')], \quad (9)$$

where $u_p(\mathbf{x}, \mathbf{x}') = \nabla_{\mathbf{x}} \log p(\mathbf{x})^\top k(\mathbf{x}, \mathbf{x}')\nabla_{\mathbf{x}} \log p(\mathbf{x}') + \nabla_{\mathbf{x}} \log p(\mathbf{x})^\top \nabla_{\mathbf{x}}k(x, \mathbf{x}') + \nabla_{\mathbf{x}}k(\mathbf{x}, \mathbf{x}')^\top \nabla_{\mathbf{x}} \log p(\mathbf{x}') + tr(\nabla_{\mathbf{x}, \mathbf{x}'}k(\mathbf{x}, \mathbf{x}'))$. This (9) gives the Kernel Stein Discrepancy (KSD). An equivalent definition is

$$\mathcal{S}_K(\mathbb{Q}, \mathbb{P}) = \mathbb{E}_{\mathbf{x}, \mathbf{x}' \sim \mathbb{P}}[(\nabla_x \log d\mathbb{P}/d\mathbb{Q}(\mathbf{x}))^\top k(\mathbf{x}, \mathbf{x}')\nabla_x \log d\mathbb{P}/d\mathbb{Q}(\mathbf{x}')],$$

# C  Proofs of Results in Section 4.1

## C.1  Proof of Theorem 1

*Proof.* Applying Kantorovich's duality on $\mathcal{W}(\mathbb{P}_G, \mathbb{P}_{\text{real}})$ and using the exhaustiveness assumption on the generator, we rewrite the problem as

$$\min_{E,\mathbb{P}} \max_{D} \{\mathbb{E}_{\mathbb{P}}[D] - \mathbb{E}_{\mathbb{P}_{\text{real}}}[D] + \lambda_1 \mathcal{S}(\mathbb{P}_{\text{real}}, \mathbb{P}_E) + \lambda_2 \mathcal{S}(\mathbb{P}, \mathbb{P}_E)\}, \tag{10}$$

where the minimization with respect to $E$ is over all energy functions, the minimization with respect to $\mathbb{P}$ is over all probability distributions with continuous density, and the maximization with respect to $D$ is over all 1-Lipschitz continuous functions. Recall the definition of kernel Stein discrepancy

$$\mathcal{S}(\mathbb{P}, \mathbb{P}_E) = \mathbb{E}_{\mathbf{x},\mathbf{x}'\sim\mathbb{P}}[(\nabla_x \log d\mathbb{P}/d\mathbb{P}_E(\mathbf{x}))^\top k(\mathbf{x}, \mathbf{x}')\nabla_x \log d\mathbb{P}/d\mathbb{P}_E(\mathbf{x}')],$$

where $d\mathbb{P}/d\mathbb{P}_E$ is the Radon-Nikodym derivative. Observe that $\mathcal{S}(\mathbb{P}, \mathbb{P}_E)$ is infinite if $\mathbb{P}$ is not absolutely continuous with respect to $\mathbb{P}_E$. Hence, to minimize the objective of (10), it suffices to consider those $\mathbb{P}$'s that are absolutely continuous with respect to $\mathbb{P}_E$.

Fixing $E$, we claim that we can swap $\min_h$ and $\max_D$ in (10). Indeed, introducing a change of variable $H(\mathbf{x}) = \log d\mathbb{P}/d\mathbb{P}_E$, then problem (10) becomes

$$\min_{E,h} \max_{D} \left\{ \mathbb{E}_{\mathbb{P}_E}[e^H D] - \mathbb{E}_{\mathbb{P}_{real}}[D] + \lambda_1 \mathcal{S}(\mathbb{P}_{real}, \mathbb{P}_E) + \lambda_2 \mathbb{E}_{\mathbf{x},\mathbf{x}'\sim\mathbb{P}}[\nabla_x H(\mathbf{x})^\top k(\mathbf{x}, \mathbf{x}')\nabla_x H(\mathbf{x}')] \right\}.$$

The objective function is linear in $D$ and convex in $H$ due to the convexity of the exponential function, the linearity of expectation operator and differential operator, and the positive definiteness of $k$. Without loss of generality, we can restrict $D$ to be such that $D(\mathbf{x}_0) = 0$ for some element $\mathbf{x}_0$, as a constant shift does not change the value of $\mathbb{E}_{\mathbb{P}_E}[(1+h)D] - \mathbb{E}_{\mathbb{P}_{\text{real}}}[D]$. The set of Lipschitz functions that vanish at $\mathbf{x}_0$ is a Banach space, and the set of 1-Lipschitz functions is compact [44]. Moreover, $L^1(\mathbb{P}_E)$ is also a Banach space and the objective function is linear in both $h$ and $D$. The above verifies the condition of Sion's minimax theorem, and thus the claim is proved.

Swapping $\min_h$ and $\max_D$ in (11). Introducing a variable replacement $h := e^H - 1 = d\mathbb{P}/d\mathbb{P}_E - 1$, then problem (10) becomes

$$\min_{E} \max_{D} \min_{h} \left\{ \mathbb{E}_{\mathbb{P}_E}[(1+h)D] - \mathbb{E}_{\mathbb{P}_{\text{real}}}[D] + \lambda_1 \mathcal{S}(\mathbb{P}_{\text{real}}, \mathbb{P}_E) \right.$$
$$\left. + \lambda_2 \cdot \mathbb{E}_{\mathbf{x},\mathbf{x}'\sim\mathbb{P}}[\nabla_x \log(1 + h(\mathbf{x}))^\top k(\mathbf{x}, \mathbf{x}')\nabla_x \log(1 + h(\mathbf{x}'))] \right\}, \tag{11}$$

where the minimization with respect to $h$ is over all $L^1(\mathbb{P}_E)$ functions with $\mathbb{P}_E$-expectation zero. Fixing $E$ and $D$, we consider

$$\min_{h:\mathbb{E}_{\mathbb{P}_E}[h]=0} \{\mathbb{E}_{\mathbb{P}_E}[hD] + \lambda_2 \cdot \mathbb{E}_{\mathbf{x},\mathbf{x}'\sim\mathbb{P}}[\nabla_x \log(1 + h(\mathbf{x}))^\top k(\mathbf{x}, \mathbf{x}')\nabla_x \log(1 + h(\mathbf{x}'))]\}$$

$$= \min_{h:\mathbb{E}_{\mathbb{P}_E}[h]=0} \left\{ \mathbb{E}_{\mathbb{P}_E}[hD] + \lambda_2 \cdot \mathbb{E}_{\mathbf{x},\mathbf{x}'\sim\mathbb{P}} \left[ \frac{\nabla_x h(\mathbf{x})^\top}{1 + h(\mathbf{x})} k(\mathbf{x}, \mathbf{x}') \frac{\nabla_x h(\mathbf{x}')}{1 + h(\mathbf{x}')} \right] \right\}$$

$$= \min_{h:\mathbb{E}_{\mathbb{P}_E}[h]=0} \left\{ \mathbb{E}_{\mathbb{P}_E}[hD] + \lambda_2 \cdot \mathbb{E}_{\mathbf{x},\mathbf{x}'\sim\mathbb{P}_E} \left[ \nabla_x h(\mathbf{x})^\top k(\mathbf{x}, \mathbf{x}')\nabla_x h(\mathbf{x}') \right] \right\},$$

where the first equality follows from the chain rule of the derivative, and the second equality follows from a change of measure $d\mathbb{P} = (1 + h)d\mathbb{P}_E$. Introducing an auxiliary variable $r$ so that $r^2$ is an upper bound of $\mathbb{E}_{\mathbf{x},\mathbf{x}'\sim\mathbb{P}_E} \left[ \nabla_x h(\mathbf{x})^\top k(\mathbf{x}, \mathbf{x}')\nabla_x h(\mathbf{x}') \right]$, we have that

$$\min_{h:\mathbb{E}_{\mathbb{P}_E}[h]=0} \left\{ \mathbb{E}_{\mathbb{P}_E}[hD] + \lambda_2 \cdot \mathbb{E}_{\mathbf{x},\mathbf{x}'\sim\mathbb{P}_E} \left[ \nabla_x h(\mathbf{x})^\top k(\mathbf{x}, \mathbf{x}')\nabla_x h(\mathbf{x}') \right] \right\}$$

$$= \min_{r\geq 0} \min_{h:\mathbb{E}_{\mathbb{P}_E}[h]=0} \left\{ \mathbb{E}_{\mathbb{P}_E}[hD] + \lambda_2 r^2 : \mathbb{E}_{\mathbf{x},\mathbf{x}'\sim\mathbb{P}_E} \left[ \nabla_x h(\mathbf{x})^\top k(\mathbf{x}, \mathbf{x}')\nabla_x h(\mathbf{x}') \right] \leq r^2 \right\}$$

$$= \min_{r\geq 0} \min_{h:\mathbb{E}_{\mathbb{P}_E}[h]=0} \left\{ r\mathbb{E}_{\mathbb{P}_E}[hD] + \lambda_2 r^2 : \mathbb{E}_{\mathbf{x},\mathbf{x}'\sim\mathbb{P}_E} \left[ \nabla_x h(\mathbf{x})^\top k(\mathbf{x}, \mathbf{x}')\nabla_x h(\mathbf{x}') \right] \leq 1 \right\}$$

$$= \min_{r\geq 0} \left\{ \lambda_2 r^2 - r \|D\|_{H^{-1}(\mathbb{P}_E;k)} \right\}$$

$$= -\frac{1}{4\lambda_2} \|D\|^2_{H^{-1}(\mathbb{P}_E;k)},$$

where the first equality holds because the minimization over $r$ forces $r^2 = \mathbb{E}_{\mathbf{x},\mathbf{x}'\sim\mathbb{P}_E}[\nabla_x h(\mathbf{x})^\top k(\mathbf{x},\mathbf{x}')\nabla_x h(\mathbf{x}')]$ at optimality; the second equality follows from a change of variable from $h$ to $rh$; and the third equality follows from the definition of the kernel Sobolev dual norm. Plugging back in (11) yields the ideal result. $\qquad\square$

## C.2  Proof for Theorem 2

*Proof.* Applying the definition of Stein discrepancy on $\mathcal{S}(\mathbb{P}_E,\mathbb{P}_G)$ and under the exhaustiveness assumption of $G$, we rewrite the problem as

$$\min_{E,\mathbb{P}}\max_{\mathbf{f}}\{\lambda_1\mathcal{S}(\mathbb{P}_{\text{real}},\mathbb{P}_E) + \lambda_2\mathbb{E}_{\mathbf{y}\sim\mathbb{P}}[\mathcal{A}_{\mathbb{P}_E}\mathbf{f}(\mathbf{y})] + \mathcal{W}(\mathbb{P}_{\text{real}},\mathbb{P})\},$$

where the minimization with respect to $E$ is over the set of all engergy functions; the minimization with respect to $\mathbb{P}$ is over all distributions; and the maximization with respect to $\mathbf{f}$ is over the Stein class for $\mathbb{P}_E$. Observe that by definition, $\mathbb{E}_{\mathbb{P}}[\mathcal{A}_{\mathbb{P}_E}\mathbf{f}(\mathbf{y})]$ equals $\mathbb{E}_{\mathbb{P}}[\nabla_y\log d\mathbb{P}_E/d\mathbb{P}(\mathbf{y})\mathbf{f}(\mathbf{y})^\top + \nabla_y\mathbf{f}(\mathbf{y})]$, which is infinite if $\mathbb{P}$ is not absolutely continuous in $\mathbb{P}_E$, hence those $\mathbb{P}$'s that are not absolutely continuous in $\mathbb{P}_E$ are automatically ruled out.

Let us fix $E$. Using a similar argument as in the proof of Theorem 1, it suffices to restrict $\mathbb{P}$ on the set of distributions that are absolutely continuous with respect to $\mathbb{P}_E$, which can be identified as the set of $L^1(\mathbb{P}_E)$ functions with $\mathbb{P}_E$-mean zero and is thus Banach. Together with the compactness assumption of the Stein class, using Sion's minimax theorem, we can swap the minimization over $\mathbb{P}$ and the maximization over $\mathbf{f}$. Now, fixing further $\mathbf{f}$, consider

$$\min_{\mathbb{P}}\{\lambda_2\mathbb{E}_{\mathbf{y}\sim\mathbb{P}}[\mathcal{A}_{\mathbb{P}_E}\mathbf{f}(\mathbf{y})] + \mathcal{W}(\mathbb{P}_{\text{real}},\mathbb{P})\}. \tag{12}$$

Recall the definition of Wasserstein metric

$$\mathcal{W}(\mathbb{P}_{\text{real}},\mathbb{P}) = \min_{\gamma}\mathbb{E}_{(\mathbf{x},\mathbf{y})\sim\gamma}[\|\mathbf{x}-\mathbf{y}\|],$$

where the minimization is over all joint distributions of $(\mathbf{x},\mathbf{y})$ with $\mathbf{x}$-marginal $\mathbb{P}_{\text{real}}$ and $\mathbf{y}$-marginal $\mathbb{P}$. We rewrite problem (12) as

$$\min_{\mathbb{P},\gamma}\{\mathbb{E}_{(\mathbf{x},\mathbf{y})\sim\gamma}[\lambda_2\,\mathcal{A}_{\mathbb{P}_E}\mathbf{f}(\mathbf{y}) + \|\mathbf{x}-\mathbf{y}\|]\},$$

where $\gamma$ has marginals $\mathbb{P}_{\text{real}}$ and $\mathbb{P}$. Since $\mathbb{P}$ is unconstrained, the above problem is further equivalent to

$$\min_{\gamma}\{\mathbb{E}_{(\mathbf{x},\mathbf{y})\sim\gamma}[\lambda_2\mathcal{A}_{\mathbb{P}_E}\mathbf{f}(\mathbf{y})] + \|\mathbf{x}-\mathbf{y}\|]\},$$

where the minimization is over all joint distributions of $(\mathbf{x},\mathbf{y})$ with $\mathbf{x}$-marginal being $\mathbb{P}_{\text{real}}$. Using the law of total expectation, the problem above is equivalent to

$$\min_{\{\gamma_{\mathbf{x}}\}_{\mathbf{x}\in\text{supp}\mathbb{P}_{\text{real}}}}\mathbb{E}_{\mathbf{x}\sim\mathbb{P}_{\text{real}}}\Big[\mathbb{E}_{\mathbf{y}\sim\gamma_{\mathbf{x}}}[\lambda_2\mathcal{A}_{\mathbb{P}_E}\mathbf{f}(\mathbf{y}) + \|\mathbf{x}-\mathbf{y}\| \mid \mathbf{x}]\Big]$$
$$= \mathbb{E}_{\mathbf{x}\sim\mathbb{P}_{\text{real}}}\left[\min_{\gamma_{\mathbf{x}}}\Big\{\mathbb{E}_{\mathbf{y}\sim\gamma_{\mathbf{x}}}[\lambda_2\mathcal{A}_{\mathbb{P}_E}\mathbf{f}(\mathbf{y}) + \|\mathbf{x}-\mathbf{y}\| \mid \mathbf{x}]\Big\}\right] \quad ,$$
$$= \mathbb{E}_{\mathbf{x}\sim\mathbb{P}_{\text{real}}}\left[\min_{\mathbf{y}\in\mathcal{X}}\{\lambda_2\mathcal{A}_{\mathbb{P}_E}\mathbf{f}(\mathbf{y}) + \|\mathbf{x}-\mathbf{y}\|\}\right]$$

where the minimization in the first line of the equation is over $\gamma_{\mathbf{x}}$, the set of all conditional distributions of $\mathbf{y}$ given $\mathbf{x}$ where $\mathbf{x}$ is over the support $\text{supp}\,\mathbb{P}_{\text{real}}$ of $\mathbb{P}_{\text{real}}$; the exchanging of $\min$ and $\mathbb{E}$ in the first equality follows from the interchangebability principle [40]; the second equality holds because the infimum can be restricted to the set of point masses. This is because the inner minimization over $\gamma_{\mathbf{x}}$ in the second line above can be attained at a Dirac mass concentrated on the minimizer $\arg\min_{\mathbf{y}}\{\lambda_2\mathcal{A}_{\mathbb{P}_E}\mathbf{f}(\mathbf{y}) + \|\mathbf{x}-\mathbf{y}\|\}$, provided that the minimizer exists; otherwise we can use an approximation argument to show it suffices to only consider point masses. Finally, the original problem is equivalent to

$$\min_{E}\max_{\mathbf{f}}\left\{\lambda_1\mathcal{S}(\mathbb{P}_{\text{real}},\mathbb{P}_E) + \mathbb{E}_{\mathbf{x}\sim\mathbb{P}_{\text{real}}}\left[\min_{\mathbf{y}\in\mathcal{X}}\{\lambda_2\mathcal{A}_{\mathbb{P}_E}\mathbf{f}(\mathbf{y}) + \|\mathbf{x}-\mathbf{y}\|\}\right]\right\}.$$

Therefore, the proof is completed using the definition of Moreau-Yosida regularization. $\qquad\square$

# D Details and Proofs in Section 4.2

## D.1 One-Dimensional Case

**Proposition 2.** *Using alternate SGD for (6) geometrically decreases the square norm $N_t = |\psi^t|^2 + |\theta - 1|^2 + |\phi + 1|^2$, for any $0 < \eta < 1$ with $\lambda_1 = \lambda_2 = 1$,*

$$N_{t+1} = (1 - \eta^2(1 - \eta)^2)N_t. \tag{13}$$

*Proof.* Instead of directly studying the optimization for (6), we first prove the following problem will converge to the unique optimum,

$$\min_\theta \max_\psi \min_\phi \theta\psi + \theta\phi + \frac{1}{2}\theta^2 + \phi^2. \tag{14}$$

Applying alternate SGD we have the following iterations:

$$\psi_{t+1} = \psi_t + \eta * \theta_t,$$

$$\phi_{t+1} = \phi_t - \eta * (\theta_t + 2\phi_t) = (1 - 2\eta)\phi_t - \eta\theta_t,$$

$$\theta_{t+1} = \theta_t - \eta(\psi_{t+1} + \phi_{t+1} + \theta_t) = -\eta(1 - 2\eta)\phi_t + (1 - \eta)\theta_t - \eta\psi_t.$$

Then we obtain the relationship between adjacent iterations:

$$\begin{bmatrix} \psi_{t+1} \\ \phi_{t+1} \\ \theta_{t+1} \end{bmatrix} = \begin{bmatrix} 1 & 0 & \eta \\ 0 & 1 - 2\eta & -\eta \\ -\eta & -\eta(1 - 2\eta) & 1 - \eta \end{bmatrix} \cdot \begin{bmatrix} \psi_t \\ \phi_t \\ \theta_t \end{bmatrix} = M \cdot \begin{bmatrix} \psi_t \\ \phi_t \\ \theta_t \end{bmatrix}$$

We further calculate the eigenvalues for matrix $M$ and have the following equations (assume the eigenvalue as $\lambda$):

$$(\lambda - 1)^3 + 3\eta(\lambda - 1)^2 + 2\eta^2(1 + \eta)(\lambda - 1) + 2\eta^3 = 0.$$

One can verify that the solutions to the above equation satisfy $|\lambda| < \sqrt{(1 - \eta + \eta^2)(1 + \eta - \eta^2)}$.

Then we have the following relationship

$$\left\| \begin{bmatrix} \psi_{t+1} \\ \phi_{t+1} \\ \theta_{t+1} \end{bmatrix} \right\|_2^2 = \left\| [\psi_t \quad \phi_t \quad \theta_t] \cdot M^\top M \cdot \begin{bmatrix} \psi_t \\ \phi_t \\ \theta_t \end{bmatrix} \right\|_2^2 \leq \lambda_m^2 \cdot \left\| \begin{bmatrix} \psi_t \\ \phi_t \\ \theta_t \end{bmatrix} \right\|_2^2$$

where $\lambda_m$ denotes the eigenvalue with the maximum absolute value of matrix $M$. Hence, we have

$$\psi_{t+1}^2 + \phi_{t+1}^2 + \theta_{t+1}^2 \leq (1 - \eta + \eta^2)(1 + \eta - \eta^2)[\psi_t^2 + \phi_t^2 + \theta_t^2].$$

We proceed to replace $\psi$, $\phi$ and $\theta$ in (14) by $\psi'$, $\phi'$ and $\theta'$ respectively and conduct a change of variable: let $\theta' = 1 - \theta$ and $\phi' = -1 - \phi$. Then we get the conclusion in the proposition.

$\square$

As shown in Fig. 3(b), Stein Bridging achieves a good convergence to the right optimum. Compared with (4), the objective (6) adds a new bilinear term $\phi \cdot \theta$, which acts like a connection between the generator and estimator, and two other quadratic terms, which help to penalize the increasing of values through training. The added terms and original terms in (6) cooperate to guarantee convergence to a unique optimum. In fact, the added terms $\frac{\lambda_1}{2}(1 + \phi)^2 + \frac{\lambda_2}{2}(\theta + \phi)^2$ in (6) and the original terms $\psi - \psi \cdot \theta$ in WGAN play both necessary roles to guarantee the convergence to the unique optimum points $[\psi^*, \theta^*, \phi^*] = [0, 1, -1]$. If we remove the critic and optimize $\theta$ and $\phi$ with the remaining loss terms, we would find that the training would converge but not necessarily to $[\psi^*, \theta^*] = [0, 1]$ (since the optimum points are not unique in this case). On the other hand, if we remove the estimator, the system degrades to (4) and would not converge to the unique optimum point $[\psi^*, \theta^*] = [0, 1]$. If we consider both of the world and optimize three terms together, the training would converge to a unique global optimum $[\psi^*, \theta^*, \phi^*] = [0, 1, -1]$.

## D.2 Generalization to Bilinear Systems

Our analysis in the one-dimension case inspires us that we can add affiliated variable to modify the objective and stabilize the training for general bilinear system. The bilinear system is of wide interest for researchers focusing on stability of GAN training ([13, 26, 12, 11, 51]). The general bilinear function can be written as

$$F(\psi, \theta) = \theta^\top \mathbf{A} \psi - \mathbf{b}^\top \theta - \mathbf{c}^\top \psi, \tag{15}$$

where $\psi, \theta$ are both $r$-dimensional vectors and the objective is $\min_{\theta} \max_{\psi} F(\psi, \theta)$ which can be seen as a basic form of various GAN objectives. Unfortunately, if we directly use simultaneous (resp. alternate) SGD to optimize such objectives, one can obtain divergence (resp. fluctuation). To solve the issue, some recent papers propose several optimization algorithms, like extrapolation from the past ([12]), crossing the curl ([11]) and consensus optimization ([26]). Also, [26] shows that it is the interaction term which generates non-zero values for $\nabla_{\theta\psi} F$ and $\nabla_{\psi\theta} F$ that leads to such instability of training. Different from previous works that focused on algorithmic perspective, we propose to add new affiliated variables which modify the objective function and allow the SGD algorithm to achieve convergence without changing the optimum points.

Based on the minimax objective of (15) we add affiliated $r$-dimensional variable $\phi$ (corresponding to the estimator in our model) the original system and tackle the following problem:

$$\min_{\theta} \max_{\psi} \min_{\phi} F(\psi, \theta) + \alpha H(\phi, \theta), \tag{16}$$

where $H(\phi, \theta) = \frac{1}{2}(\theta + \phi)^\top \mathbf{B}(\theta + \phi)$, $\mathbf{B} = (\mathbf{A}\mathbf{A}^\top)^{\frac{1}{2}}$ and $\alpha$ is a non-negative constant. Theoretically, the new problem keeps the optimum points of (15) unchanged. Let $L(\psi, \phi, \theta) = F(\psi, \theta) + \alpha G(\phi, \theta)$.

**Proposition 3.** *Assume the optimum point of $\min_{\theta} \max_{\psi} F(\psi, \theta)$ are $[\psi^*, \theta^*]$, then the optimum points of (16) would be $[\psi^*, \theta^*, \phi^*]$ where $\phi^* = -\theta^*$.*

*Proof.* The condition tells us that $\nabla_{\theta} F(\psi^*, \theta) = 0$ and $\nabla_{\psi} F(\psi, \theta^*) = 0$. Then we derive the gradients for $L(\psi, \phi, \theta)$,

$$\nabla_{\psi} L(\psi^*, \phi, \theta) = \nabla_{\theta} F(\psi^*, \theta) = 0, \tag{17}$$

$$\nabla_{\theta} L(\psi, \phi, \theta^*) = \nabla_{\theta} F(\psi, \theta^*) + \nabla_{\theta} H(\phi, \theta^*) = \frac{1}{2}(\mathbf{B} + \mathbf{B}^\top)(\theta^* + \phi), \tag{18}$$

$$\nabla_{\phi} L(\psi, \phi, \theta) = \nabla_{\phi} H(\phi, \theta) = \frac{1}{2}(\mathbf{B} + \mathbf{B}^\top)(\phi + \theta), \tag{19}$$

Combining (18) and (19) we get $\phi^* = -\theta^*$. Hence, the optimum point of (16) is $[\psi^*, \theta^*, \phi^*]$ where $\phi^* = -\theta^*$. $\qquad\square$

The advantage of the new problem is that it can be solved by SGD algorithm and guarantees convergence theoretically. We formulate the results in the following theorem.

**Theorem 3.** *For problem $\min_{\theta} \max_{\psi} \min_{\phi} L(\psi, \phi, \theta)$ using alternate SGD algorithm, i.e.,*

$$\begin{aligned}
\psi_{t+1} &= \psi_t + \eta \nabla_{\psi} L(\theta_t, \psi_t, \phi_t), \\
\phi_{t+1} &= \phi_t - \eta \nabla_{\phi} L(\theta_t, \psi_{t+1}, \phi_t), \\
\theta_{t+1} &= \theta_t - \eta \nabla_{\theta} L(\theta_t, \psi_{t+1}, \phi_{t+1}),
\end{aligned} \tag{20}$$

*we can achieve convergence to $[\psi^*, \theta^*, \phi^*]$ where $\phi^* = -\theta^*$ with at least linear rate of $(1 - \eta_1 + \eta_2^2)(1 + \eta_2 - \eta_1^2)$ where $\eta_1 = \eta\sigma_{min}$, $\eta_2 = \eta\sigma_{max}$ and $\sigma_{min}$ (resp. $\sigma_{max}$) denotes the maximum (resp. minimum) singular value of matrix $\mathbf{A}$.*

To prove Theorem 3, we can prove a more general argument.

**Proposition 1.** *If we consider any first-order optimization method on (16), i.e.,*

$$\psi_{t+1} \in \psi_0 + span(L(\psi_0, \phi, \theta), \cdots, F(\psi_t, \phi, \theta)), \forall t \in \mathbb{N},$$

$$\phi_{t+1} \in \psi_0 + span(L(\psi, \phi_0, \theta), \cdots, L(\psi, \phi_t, \theta)), \forall t \in \mathbb{N},$$

$$\boldsymbol{\theta}_{t+1} \in \boldsymbol{\psi}_0 + span(L(\boldsymbol{\psi}, \boldsymbol{\phi}, \boldsymbol{\theta}_0), \cdots, L(\boldsymbol{\psi}, \boldsymbol{\phi}, \boldsymbol{\theta}_t)), \forall t \in \mathbb{N},$$

*Then we have*

$$\widetilde{\boldsymbol{\psi}}_t = \mathbf{V}^\top(\boldsymbol{\psi}_t - \boldsymbol{\psi}^*), \quad \widetilde{\boldsymbol{\phi}}_t = \mathbf{U}^\top(\boldsymbol{\phi}_t - \boldsymbol{\phi}^*), \quad \widetilde{\boldsymbol{\theta}}_t = \mathbf{U}^\top(\boldsymbol{\theta}_t - \boldsymbol{\theta}^*),$$

*where $\mathbf{U}$ and $\mathbf{V}$ are the singular vectors decomposed by matrix $\mathbf{A}$ using SVD decomposition, i.e., $\mathbf{A} = \mathbf{U}\mathbf{D}\mathbf{V}^\top$ and the triple $([\widetilde{\boldsymbol{\psi}}_t]_i, [\widetilde{\boldsymbol{\phi}}_t]_i, [\widetilde{\boldsymbol{\theta}}_t]_i)_{1 \le i \le r}$ follows the update rule with step size $\sigma_i \eta$ as the same optimization method on a unidimensional problem*

$$\min_\theta \max_\psi \min_\phi \theta\psi + \theta\phi + \frac{1}{2}\theta^2 + \frac{1}{2}\phi^2, \tag{21}$$

*with step size $\eta$, where $\sigma_i$ denotes the $i$-th singular value on the diagonal of $\mathbf{D}$.*

*Proof.* The proof is extended from the proof of Lemma 3 in [12]. The general class of first-order optimization methods derive the following updations:

$$\boldsymbol{\psi}_{t+1} = \boldsymbol{\psi}_0 + \sum_{s=0}^{t+1} \rho_{st}(\mathbf{A}^\top\boldsymbol{\theta}_s - \mathbf{c}) = \boldsymbol{\psi}_0 + \sum_{s=0}^{t+1} \rho_{st}\mathbf{A}^\top(\boldsymbol{\theta}_s - \boldsymbol{\theta}^*),$$

$$\boldsymbol{\phi}_{t+1} = \boldsymbol{\phi}_0 + \frac{1}{2}\sum_{s=0}^{t+1} \delta_{st}(\mathbf{B} + \mathbf{B}^\top)(\boldsymbol{\theta}_s + \boldsymbol{\phi}_s),$$

$$\boldsymbol{\theta}_{t+1} = \boldsymbol{\theta}_0 + \sum_{s=0}^{t+1} \mu_{st}[\mathbf{A}(\boldsymbol{\psi}_s - \boldsymbol{\psi}^*) + \frac{1}{2}(\mathbf{B} + \mathbf{B}^\top)(\boldsymbol{\theta}_s + \boldsymbol{\phi}_s)],$$

where $\rho_{st}, \delta_{st}, \mu_{st} \in \mathbb{R}$ depend on specific optimization method (for example, in SGD, $\rho_{tt} = \delta_{tt} = \mu_{tt}$ remain as a non-zero constant for $\forall t$ and other coefficients are zero).

Using SVD $\mathbf{A} = \mathbf{U}\mathbf{D}\mathbf{V}^\top$ and the fact $\theta^* = -\phi^*$, $\mathbf{B} = (\mathbf{U}\mathbf{D}\mathbf{D}^\top\mathbf{U}^\top) = \mathbf{D}$, we have

$$\mathbf{V}^\top(\boldsymbol{\psi}_{t+1} - \boldsymbol{\psi}^*) = \mathbf{V}^\top(\boldsymbol{\psi}_0 - \boldsymbol{\psi}^*) + \sum_{s=0}^{t+1} \rho_{st}\mathbf{D}^\top\mathbf{U}^\top(\boldsymbol{\theta}_s - \boldsymbol{\theta}^*)$$

$$\mathbf{U}^\top(\boldsymbol{\phi}_{t+1} - \boldsymbol{\phi}^*) = \mathbf{U}^\top(\boldsymbol{\phi}_0 - \boldsymbol{\phi}^*) + \sum_{s=0}^{t+1} \delta_{st}\mathbf{U}^\top\mathbf{D}(\boldsymbol{\theta}_s - \boldsymbol{\theta}^*) + \mathbf{U}^\top\mathbf{D}(\boldsymbol{\phi}_s - \boldsymbol{\phi}^*),$$

$$\mathbf{U}^\top(\boldsymbol{\theta}_{t+1} - \boldsymbol{\theta}^*) = \mathbf{U}^\top(\boldsymbol{\theta}_0 - \boldsymbol{\theta}^*) + \sum_{s=0}^{t+1} \rho_{st}[\mathbf{D}\mathbf{V}^\top(\boldsymbol{\psi}_s - \boldsymbol{\psi}^*) + \mathbf{U}^\top\mathbf{D}(\boldsymbol{\theta}_s - \boldsymbol{\theta}^*) + \mathbf{U}^\top\mathbf{D}(\boldsymbol{\phi}_s - \boldsymbol{\phi}^*)],$$

and equivalently,

$$\widetilde{\boldsymbol{\psi}}_{t+1} = \widetilde{\boldsymbol{\psi}}_0 + \sum_{s=0}^{t+1} \rho_{st}\mathbf{D}^\top\widetilde{\boldsymbol{\theta}}_t, \quad \widetilde{\boldsymbol{\phi}}_t = \widetilde{\boldsymbol{\phi}}_0 + \sum_{s=0}^{t+1} \delta_{st}\mathbf{D}(\widetilde{\boldsymbol{\theta}}_t + \widetilde{\boldsymbol{\phi}}_t),$$

$$\widetilde{\boldsymbol{\theta}}_{t+1} = \widetilde{\boldsymbol{\theta}}_0 + \sum_{s=0}^{t+1} \rho_{st}\mathbf{D}(\widetilde{\boldsymbol{\psi}}_t + \widetilde{\boldsymbol{\theta}}_t + \widetilde{\boldsymbol{\phi}}_t).$$

Note that $\mathbf{D}$ is a rectangular matrix with non-zero elements on a diagonal block of size $r$. Hence, the above $r$-dimensional problem can be reduced to $r$ unidimensional problems:

$$[\widetilde{\boldsymbol{\psi}}_{t+1}]_i = [\widetilde{\boldsymbol{\psi}}_0]_i + \sum_{s=0}^{t+1} \rho_{st}\sigma_i[\widetilde{\boldsymbol{\theta}}_t]_i, \quad [\widetilde{\boldsymbol{\phi}}_t]_i = [\widetilde{\boldsymbol{\phi}}_0]_i + \sum_{s=0}^{t+1} \delta_{st}\sigma_i([\widetilde{\boldsymbol{\theta}}_t]_i + [\widetilde{\boldsymbol{\phi}}_t]_i),$$

$$[\widetilde{\boldsymbol{\theta}}_{t+1}]_i = [\widetilde{\boldsymbol{\theta}}_0]_i + \sum_{s=0}^{t+1} \rho_{st}\sigma_i([\widetilde{\boldsymbol{\psi}}_t]_i + [\widetilde{\boldsymbol{\theta}}_t]_i + [\widetilde{\boldsymbol{\phi}}_t]_i).$$

The above iterations can be conducted independently in each dimension where the optimization in $i$-th dimension follows the same updating rule with step size $\sigma_i \eta$ as problem in (21). $\qquad \square$

Furthermore, since problem ([21]) can achieve convergence with a linear rate of $(1-\eta+\eta^2)(1+\eta-\eta^2)$ using alternate SGD (the proof is similar to that of ([14])), the multi-dimensional problem in ([16]) can achieve convergence by SGD with at least a rate of $(1-\eta_1+\eta_2^2)(1+\eta_2-\eta_1^2)$ where $\eta_1 = \eta\sigma_{max}$, $\eta_2 = \eta\sigma_{min}$ and $\sigma_{max}$ (resp. $\sigma_{min}$) denotes the maximum (resp. minimum) singular value of matrix $\mathbf{A}$. We conclude the proof for Theorem 4.

Theorem [3] suggests that the added term $H(\boldsymbol{\phi}, \boldsymbol{\theta})$ with affiliated variables $\phi$ could help the SGD algorithm achieve convergence to the the same optimum points as directly optimizing $F(\boldsymbol{\psi}, \boldsymbol{\theta})$. Our method is related to consensus optimization algorithm ([26]) which adds a regularization term $\|\nabla_{\boldsymbol{\theta}} F(\boldsymbol{\psi}, \boldsymbol{\theta})\| + \|\nabla_{\boldsymbol{\psi}} F(\boldsymbol{\psi}, \boldsymbol{\theta})\|$ to ([15]) resulting extra quadratic terms for $\boldsymbol{\theta}$ and $\boldsymbol{\psi}$. The disadvantage of such method is the requirement of Hessian matrix of $F(\boldsymbol{\psi}, \boldsymbol{\theta})$ which is computational expensive for high-dimensional data. By contrast, our solution only requires the first-order derivatives.

# E  Details for Implementations

## E.1  Synthetic Datasets

We provide the details for two synthetic datasets. The Two-Circle dataset consists of 24 Gaussian mixtures where 8 of them are located in an inner circle with radius $r_1 = 4$ and 16 of them lie in an outer circle with radius $r_2 = 8$. For each Gaussian component, the covariance matrix is $\begin{pmatrix} 0.2 & 0 \\ 0 & 0.2 \end{pmatrix} = \sigma_1 \mathbf{I}$ and the mean value is $[r_1 \cos t, r_1 \sin t]$, where $t = \frac{2\pi \cdot k}{8}$, $k = 1, \cdots, 8$, for the inner circle, and $[r_2 \cos t, r_2 \sin t]$, where $t = \frac{2\pi \cdot k}{16}$, $k = 1, \cdots, 16$ for the outer circle. We sample $N_1 = 2000$ points as true observed samples for model training.

The Two-Spiral dataset contains 100 Gaussian mixtures whose centers locate on two spiral-shaped curves. For each Gaussian component, the covariance matrix is $\begin{pmatrix} 0.5 & 0 \\ 0 & 0.5 \end{pmatrix} = \sigma_2 \mathbf{I}$ and the mean value is $[-c_1 \cos c_1, c_1 \sin c_1]$, where $c_1 = \frac{2\pi}{3} + linspace(0, 0.5, 50) \cdot 2\pi$, for one spiral, and $[c_2 \cos c_2, -c_2 \sin c_2]$, where $c_2 = \frac{2\pi}{3} + linspace(0, 0.5, 50) \cdot 2\pi$ for another spiral. We sample $N_2 = 5000$ points as true observed samples.

## E.2  Model Specifications and Training Algorithm

In different tasks, we consider different model specifications in order to meet the demand of capacify as well as test the effectiveness under various settings. Our proposed framework ([3]) adopts Wasserstein distance for the first term and two Stein discrepancies for the second and the third terms. We can write ([3]) as a more general form

$$\min_{\theta, \phi} \mathcal{D}_1(\mathbb{P}_{\text{real}}, \mathbb{P}_G) + \lambda_1 \mathcal{D}_2(\mathbb{P}_{\text{real}}, \mathbb{P}_E) + \lambda_2 \mathcal{D}_3(\mathbb{P}_G, \mathbb{P}_E), \tag{22}$$

where $\mathcal{D}_1, \mathcal{D}_2, \mathcal{D}_3$ denote three general discrepancy measures for distributions. As stated in our remark, $\mathcal{D}_1$ can be specified as arbitrary discrepancy measures for implicit generative models. Here we also use JS divergence, the objective for valina GAN. To well distinguish them, we call the model using Wasserstein distance (resp. JS divergence) as Joint-W (resp. Joint-JS) in our experiments. On the other hand, the two Stein discrepancies in ([3]) can be specified by KSD (as defined by $\mathcal{S}_k$ in ([9])) or general Stein discrepancy with an extra critic (as defined by $\mathcal{S}$ in ([1])). Hence, the two specifications for $\mathcal{D}_1$ and the two for $\mathcal{D}_2$ ($\mathcal{D}_3$) compose four different combinations in total, and we organize the objectives in each case in Table [4].

In our experiments, we use KSD with RBF kernels for $\mathcal{D}_2$ and $\mathcal{D}_3$ in Joint-W and Joint-JS on two synthetic datasets. For MNIST with conditional training (given the digit class as model input), we also use KSD with RBF kernels. For MNIST and CIFAR with unconditional training (the class is not given as known information), we find that KSD cannot provide desirable results so we adopt general Stein discrepancy for higher model capacity.

The objectives in Table [4] appear to be comutationally expensive. In the worst case (using general Stein discrepancy), there are two minimax operations where one is from GAN or WGAN and one is from Stein discrepancy estimation. To guarantee training efficiency, we alternatively update the generator, estimator, Wasserstein critic and Stein critic over the parameters $\theta$, $\phi$, $\psi$ and $\pi$ respectively.

Table 4: Objectives for different specifications of $\mathcal{D}_1(\mathbb{P}_{\text{real}}, \mathbb{P}_G)$, $\mathcal{D}_2(\mathbb{P}_{\text{real}}, \mathbb{P}_E)$ and $\mathcal{D}_3(\mathbb{P}_G, \mathbb{P}_E)$. We specify $\mathcal{D}_1$ as Wasserstein distance or JS divergence in our paper and for $\mathcal{D}_2$ and $\mathcal{D}_3$ we consider the general Stein discrepancy or kernel Stein discrepancy. Here we use $\mathcal{W}$, $\mathcal{JS}$ to denote Wasserstein distance and JS divergence respectively, and $\mathcal{S}$, $\mathcal{S}_k$ to represent general Stein discrepancy and kernel Stein discrepancy respectively. We omit the gradient penalty term for Wasserstein distance here but use it in experiments.

| $\mathcal{D}_1$ | $\mathcal{D}_2$ | $\mathcal{D}_3$ | Objective |
|---|---|---|---|
| $\mathcal{W}$ | $\mathcal{S}$ | $\mathcal{S}$ | $\min_\theta \min_\phi \max_\psi \max_\pi \mathbb{E}_{\mathbf{x} \sim \mathbb{P}_{data}}[d_\psi(\mathbf{x})] - \mathbb{E}_{\mathbf{z} \sim p_0}[d_\psi(G_\theta(\mathbf{z}))]$ $+\lambda_1 \mathbb{E}_{\mathbf{x} \sim \mathbb{P}_{data}}[\mathcal{A}_{p_\phi}[\mathbf{f}_\pi(\mathbf{x})]] + \lambda_2 \mathbb{E}_{\mathbf{z} \sim \backslash_0}[\mathcal{A}_{p_\phi}[\mathbf{f}_\pi(G_\theta(\mathbf{z}))]]$ |
| $\mathcal{W}$ | $\mathcal{S}_k$ | $\mathcal{S}_k$ | $\min_\theta \min_\phi \max_\psi \mathbb{E}_{\mathbf{x} \sim \mathbb{P}_{data}}[d_\psi(\mathbf{x})] - \mathbb{E}_{\mathbf{z} \sim p_0}[d_\psi(G_\theta(\mathbf{z}))]$ $+\lambda_1 \mathbb{E}_{\mathbf{x},\mathbf{x}' \sim \mathbb{P}_{data}}[u_{p_\phi}(x, x')] + \lambda_2 \mathbb{E}_{\mathbf{z},\mathbf{z}' \sim p_0}[u_{p_\phi}(G_\theta(\mathbf{z}), G_\theta(\mathbf{z}'))]$ |
| $\mathcal{JS}$ | $\mathcal{S}$ | $\mathcal{S}$ | $\min_\theta \min_\phi \max_\psi \max_\pi \mathbb{E}_{\mathbf{x} \sim \mathbb{P}_r}[\log(d_\psi(\mathbf{x}))] + \mathbb{E}_{\mathbf{z} \sim p_0}[\log(1 - d_\psi(G_\theta(\mathbf{z})))]$ $+\lambda_1 \mathbb{E}_{\mathbf{x} \sim \mathbb{P}_{data}}[\mathcal{A}_{p_\phi}[\mathbf{f}_\pi(\mathbf{x})]] + \lambda_2 \mathbb{E}_{\mathbf{z} \sim \backslash_0}[\mathcal{A}_{p_\phi}[\mathbf{f}_\pi(G_\theta(\mathbf{z}))]]$ |
| $\mathcal{JS}$ | $\mathcal{S}_k$ | $\mathcal{S}_k$ | $\min_\theta \min_\phi \max_\psi \mathbb{E}_{\mathbf{x} \sim \mathbb{P}_r}[\log(d_\psi(\mathbf{x}))] + \mathbb{E}_{\mathbf{z} \sim p_0}[\log(1 - d_\psi(G_\theta(\mathbf{z})))]$ $+\lambda_1 \mathbb{E}_{\mathbf{x},\mathbf{x}' \sim \mathbb{P}_{data}}[u_{p_\phi}(x, x')] + \lambda_2 \mathbb{E}_{\mathbf{z},\mathbf{z}' \sim p_0}[u_{p_\phi}(G_\theta(\mathbf{z}), G_\theta(\mathbf{z}'))]$ |

Specifically, in one iteration, we optimize the generator over $\theta$ and the estimator over $\phi$ with one step respectively, and then optimize the Wasserstein critic over $\psi$ with $n_d$ steps and the Stein critic over $\pi$ with $n_c$ steps. Such training approach guarantees the same time complexity order of proposed method as that of GAN or WGAN, and the training time for our model can be bounded within constant times the time for training GAN model. In our experiment, we set $n_d = n_c = 5$ and empirically find that our model Stein Bridging would be two times slower than WGAN on average. We present the training algorithm for Stein Bridging in Algorithm 1.

### E.3 Implementation Details

We give the information of network architectures and hyper-parameter settings for our model as well as each competitor in our experiments. The hyper-parameters are searched with grid search.

The energy function is often parametrized as a sum of multiple experts ([18]) and each expert can have various function forms depending on the distributions. If using sigmoid distribution, the energy function becomes (see section 2.1 in [22] for details)

$$E_\phi(\mathbf{x}) = \sum_i \log(1 + e^{-(\mathbf{W}_i n(\mathbf{x}) + b_i)}), \tag{23}$$

where $n(\mathbf{x})$ maps input $\mathbf{x}$ to a feature vector and could be specified as a deep neural network, which corresponds to deep energy model ([32])

When not using KSD, the implementation for Stein critic $\mathbf{f}$ and operation function $\phi$ in (1) has still remained an open problem. Some existing studies like [20] set $d' = 1$ in which situation $\mathbf{f}$ reduces to a scalar-function from $d$-dimension input to one-dimension scalar value. Such setting can reduce computational cost since large $d'$ could lead to heavy computation for training. Empirically, in our experiments on image dataset, we find that setting $d' = 1$ can provide similar performance to $d' = 10$ or $d' = 100$. Hence, we set $d' = 1$ in our experiment in order for efficiency. Besides, to further reduce computational cost, we let the two Stein critics share the parameters, which empirically provide better performance than two different Stein critics.

Another tricky point is how to design a proper $\Gamma$ given $d' \neq d$ where the trace operation is not applicable. One simple way is to set $\Gamma$ as some matrix norms. However, the issue is that using matrix norm would make it hard for SGD learning. The reason is that the $\Gamma$ and the expectation in (1) cannot exchange the order, in which case there is no unbiased estimation by mini-batch samples for the gradient. Here, we specify $\Gamma$ as max-pooling over different dimensions of $\mathcal{A}_{p_\phi}[\mathbf{f}_\pi(\mathbf{x})]$, i.e. the gradient would back-propagate through the dimension with largest absolute value at one time. Theoretically, such setting can guarantee the value in each dimension reduces to zero through training and we find it works well in practice.

For synthetic datasets, we set the noise dimension as 4. All the generators are specified as a three-layer fully-connected (FC) neural network with neuron size $4 - 128 - 128 - 2$, and all the Wasserstein

---
**Algorithm 1:** Training Algorithm for Stein Bridging
---
1 **REQUIRE:** observed training samples $\{\mathbf{x}\} \sim \mathbb{P}_{real}$.

2 **REQUIRE:** $\theta_0, \phi_0, \psi_0, \pi_0$, initial parameters for generator, estimator, Wasserstein critic and Stein critic models respectively. $\alpha^E = 0.0002, \beta_1^E = 0.9, \beta_2^E = 0.999$, Adam hyper-parameters for explicit models. $\alpha^I = 0.0002, \beta_1^I = 0.5, \beta_2^I = 0.999$, Adam hyper-parameters for implicit models. $\lambda_1 = 1, \lambda_2$, weights for $\mathcal{D}_2$ and $\mathcal{D}_3$ (we suggest increasing $\lambda_2$ from 0 to 1 through training). $n_d = 5, n_c = 5$ number of iterations for Wasserstein critic and Stein critic, respectively, before one iteration for generator and estimator. $B = 100$, batch size.

3 **while** *not converged* **do**

4     **for** $n = 1, \cdots, n_d$ **do**

5         Sample $B$ true samples $\{\mathbf{x}_i\}_{i=1}^B$ from $\{\mathbf{x}\}$;

6         Sample $B$ random noise $\{\mathbf{z}_i\}_{i=1}^B \sim P_0$ and obtain generated samples $\widetilde{\mathbf{x}}_\mathbf{i} = G_\theta(\mathbf{z}_i)$ ;

7         $\mathcal{L}_{dis} = \frac{1}{B} \sum_{i=1}^B d_\psi(\mathbf{x}_i) - d_\psi(\widetilde{\mathbf{x}}_\mathbf{i}) - \lambda(\|\nabla_{\hat{\mathbf{x}}_\mathbf{i}} d_\psi(\hat{\mathbf{x}}_\mathbf{i})\| - 1)^2$ // the last term is for gradient penalty in WGAN-GP where $\hat{\mathbf{x}}_\mathbf{i} = \epsilon_i \mathbf{x}_i + (1 - \epsilon_i)\widetilde{\mathbf{x}}_\mathbf{i}, \epsilon_i \sim U(0, 1)$;

8         $\psi_{k+1} \leftarrow Adam(-\mathcal{L}_{dis}, \psi_k, \alpha^I, \beta_1^I, \beta_2^I)$// update the Wasserstein critic;

9     **for** $n = 1, \cdots, n_c$ **do**

10        Sample $B$ true samples $\{\mathbf{x}_i\}_{i=1}^B$ from $\{\mathbf{x}\}$;

11       Sample $B$ random noise $\{\mathbf{z}_i\}_{i=1}^B \sim P_0$ and obtain generated samples $\widetilde{\mathbf{x}}_\mathbf{i} = G_\theta(\mathbf{z}_i)$ ;

12       $\mathcal{L}_{critic} = \frac{1}{B} \sum_{i=1}^B \lambda_1 \mathcal{A}_{p_\phi}[\mathbf{f}_\pi(\mathbf{x})] + \lambda_2 \mathcal{A}_{p_\phi}[\mathbf{f}_\pi(\widetilde{\mathbf{x}}_\mathbf{i})]$;

13       $\pi_{k+1} \leftarrow Adam(-\mathcal{L}_{critic}, \pi_k, \alpha^E, \beta_1^E, \beta_2^E)$// update the Stein critic;

14     Sample $B$ random noise $\{\mathbf{z}_i\}_{i=1}^B \sim P_0$ and obtain generated samples $\widetilde{\mathbf{x}}_\mathbf{i} = G_\theta(\mathbf{z}_i)$ ;

15     $\mathcal{L}_{est} = \frac{1}{B} \sum_{i=1}^B \lambda_1 \mathcal{A}_{p_\phi}[\mathbf{f}_\pi(\mathbf{x})] + \lambda_2 \mathcal{A}_{p_\phi}[\mathbf{f}_\pi(\widetilde{\mathbf{x}}_\mathbf{i})]$;

16     $\phi_{k+1} \leftarrow Adam(\mathcal{L}_{est}, \phi_k, \alpha^E, \beta_1^E, \beta_2^E)$// update the density estimator;

17     $\mathcal{L}_{gen} = \frac{1}{B} \sum_{i=1}^B -d_\psi(\widetilde{\mathbf{x}}_\mathbf{i}) + \lambda_2 \mathcal{A}_{p_\phi}[\mathbf{f}_\pi(\widetilde{\mathbf{x}}_\mathbf{i})]$;

18     $\theta_{k+1} \leftarrow Adam(\mathcal{L}_{gen}, \theta_k, \alpha^I, \beta_1^I, \beta_2^I)$// update the sample generator;

19 **OUTPUT:** trained sample generator $G_\theta(\mathbf{z})$ and density estimator $p_\phi(\mathbf{x})$.

---

critics (or the discriminators in JS-divergence-based GAN) are also a three-layer FC network with neuron size $2 - 128 - 128 - 1$. For the estimators, we set the expert number as 4 and the feature function $n(\mathbf{x})$ is a FC network with neuron size $2 - 128 - 128 - 4$. Then in the last layer we sum the outputs from each expert as the energy value $E(\mathbf{x})$. The activation units are searched within $[LeakyReLU, tanh, sigmoid, softplus]$. The learning rate $[1e - 6, 1e - 5, 1e - 4, 1e - 3, 1e - 2]$ and the batch size $[50, 100, 150, 200]$. The gradient penalty weight for WGAN is searched in $[0, 0.1, 1, 10, 100]$.

For MNIST dataset, we set the noise dimension as 100. All the critics/discriminators are implemented as a four-layer network where the first two layers adopt convolution operations with filter size 5 and stride $[2, 2]$ and the last two layers are FC layers. The size for each layer is $1 - 64 - 128 - 256 - 1$. All the generators are implemented as a four-layer networks where the first two layers are FC and the last two adopt deconvolution operations with filter size 5 and stride $[2, 2]$. The size for each layer is $100 - 256 - 128 - 64 - 1$. For the estimators, we consider the expert number as 128 and the feature function is the same as the Wasserstein critic except that the size of last layer is 128. Then we sum the outputs from each expert as the energy value. The activation units are searched within $[ReLU, LeakyReLU, tanh]$. The learning rate $[2e - 5, 2e - 4, 2e - 3, 2e - 2]$ and the batch size $[32, 64, 100, 128]$. The gradient penalty weight for WGAN is searched in $[1, 10, 100, 1000]$.

For CIFAR dataset, we adopt the same architecture as DCGAN for critics and generators. As for the estimator, the architecture of feature function is the same as the critics except the last year where we set the expert number as 128 and sum each output as the output energy value. The architectures for Stein critic are the same as Wasserstein critic for both MNIST and CIFAR datasets. In other words, we consider $d' = 1$ in (1) and further simply $\phi$ as an average of each dimension of $\mathbb{E}_{\mathbf{x} \sim \mathbb{P}}[\mathcal{A}_{\mathbb{Q}} \mathbf{f}(\mathbf{x})]$. Empirically we found this setting can provide efficient computation and decent performance.

### E.4 Evaluation Metrics

We adopt some quantitative metrics to evaluate the performance of each method on different tasks. In section 4.1, we use two metrics to test the sample quality: Maximum Mean Discrepancy (MMD) and High-quality Sample Rate (HSR). MMD measures the discrepancy between two distributions $X$ and $Y$, $MMD(X,Y) = \|\frac{1}{n}\sum_{i=1}^{n}\Phi(x_i) - \frac{1}{m}\sum_{j=1}^{m}\Phi(y_i)\|$ where $x_i$ and $y_j$ denote samples from $X$ and $Y$ respectively and $\Phi$ maps each sample to a RKHS. Here we use RBF kernel and calculate MMD between generated samples and true samples. HSR statistics the rate of high-quality samples over all generated samples. For Two-Cirlce dataset, we define the generated points whose distance from the nearest Gaussian component is less than $\sigma_1$ as high-quality samples. We generate 2000 points in total and statistic HSR. For Two-Spiral dataset, we set the distance threshold as $5\sigma_2$ and generate 5000 points to calculate HSR. For CIFAR, we use the Inception V3 Network in Tensorflow as pre-trained classifier to calculate inception score.

In section 4.2, we use three metrics to characterize the performance for density estimation: KL divergence, JS divergence and AUC. We divide the map into a $300$ meshgrid, calculate the unnormalized density values of each point given by the estimators and compute the KL and JS divergences between estimated density and ground-truth density. Besides, we select the centers of each Gaussian components as positive examples (expected to have high densities) and randomly sample 10 points within a circle around each center as negative examples (expected to have relatively low densities) and rank them according to the densities given by the model. Then we obtain the area under the curve (AUC) for false-positive rate v.s. true-positive rate.

Table 5: Distances between means of generated digits (resp. images) and true digits (resp. images) on MNIST (resp. CIFAR-10).

| Method | MNIST | | CIFAR | |
| --- | --- | --- | --- | --- |
| | $l_1$ Dis | $l_2$ Dis | $l_1$ Dis | $l_2$ Dis |
| WGAN-GP | 13.80 | 0.93 | 80.98 | 1.72 |
| WGAN+LR | 12.91 | 0.86 | 82.96 | 1.81 |
| WGAN+ER | 12.26 | 0.77 | 72.28 | 1.59 |
| WGAN+VA | 12.38 | 0.78 | 69.01 | 1.53 |
| DGM | 12.12 | 0.79 | 179.30 | 3.95 |
| Joint-W | **11.82** | **0.73** | **64.23** | **1.41** |

Table 6: Quantitative results including MMD (lower is better), HSR (higher is better) as the metrics for quality of generated samples and KLD (lower is better), JSD (lower is better), AUC (higher is better) as the metrics for accuracy of estimated densities on Two-Circle and Two-Spiral datasets.

| Method | Two-Cirlce | | | | | | Two-Spiral | | | | |
| --- | --- | --- | --- | --- | --- | --- | --- | --- | --- | --- | --- |
| | MMD | HSR | KLD | JSD | AUC | | MMD | HSR | KLD | JSD | AUC |
| GAN | 0.0033 | 0.772 | - | - | - | | 0.0082 | 0.583 | - | - | - |
| GAN+VA | 0.0118 | 0.295 | - | - | - | | 0.0085 | 0.761 | - | - | - |
| WGAN-GP | 0.0010 | 0.841 | - | - | - | | 0.0090 | 0.697 | - | - | - |
| WGAN+VA | 0.0016 | 0.835 | - | - | - | | 0.0159 | 0.618 | - | - | - |
| DEM | - | - | 2.036 | 0.431 | 0.683 | | - | - | 1.206 | 0.315 | 0.640 |
| EGAN | - | - | 3.350 | 0.474 | 0.616 | | - | - | 1.916 | 0.445 | 0.499 |
| DGM | 0.0040 | 0.774 | 2.272 | 0.445 | 0.600 | | 0.0019 | 0.833 | 1.725 | 0.414 | 0.589 |
| Joint-JS | 0.0037 | **0.883** | 1.104 | 0.297 | **0.962** | | 0.0031 | 0.717 | 0.655 | 0.193 | 0.808 |
| Joint-W | **0.0007** | 0.844 | **1.030** | **0.281** | 0.961 | | **0.0003** | **0.909** | **0.364** | **0.110** | **0.810** |

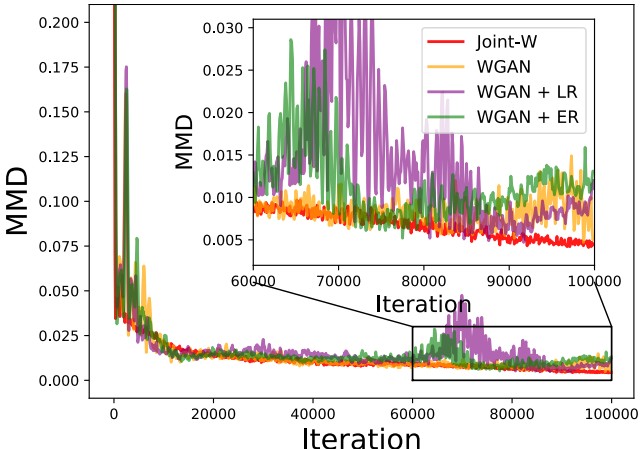

Figure 9: Learning curves in Two-Spiral dataset.

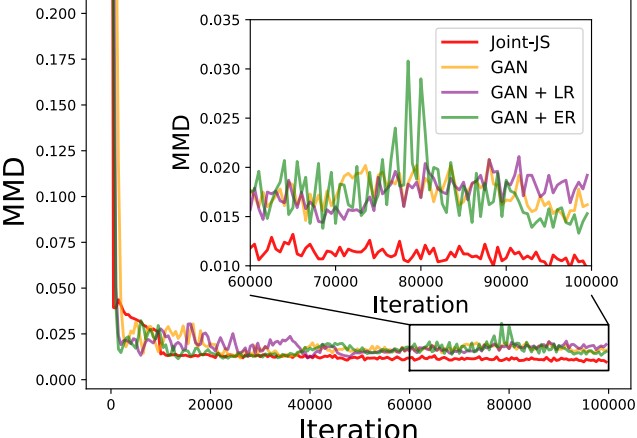

Figure 10: Learning curves in Two-Circle dataset.

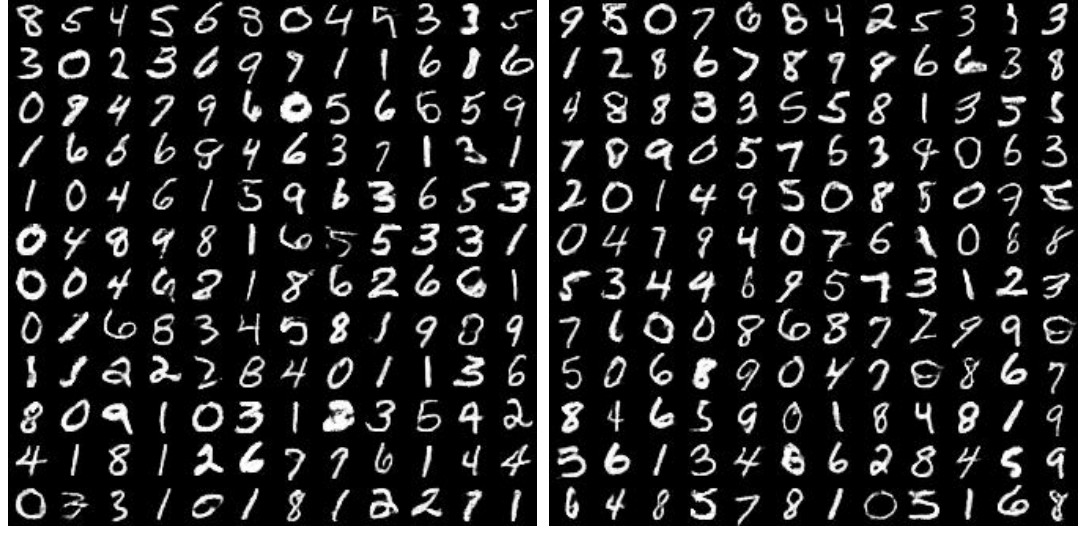

(a) Randomly sampled over all digits  (b) Randomly sampled over digits with top 50% densities

Figure 11: Generated digits given by Joint-W on MNIST.

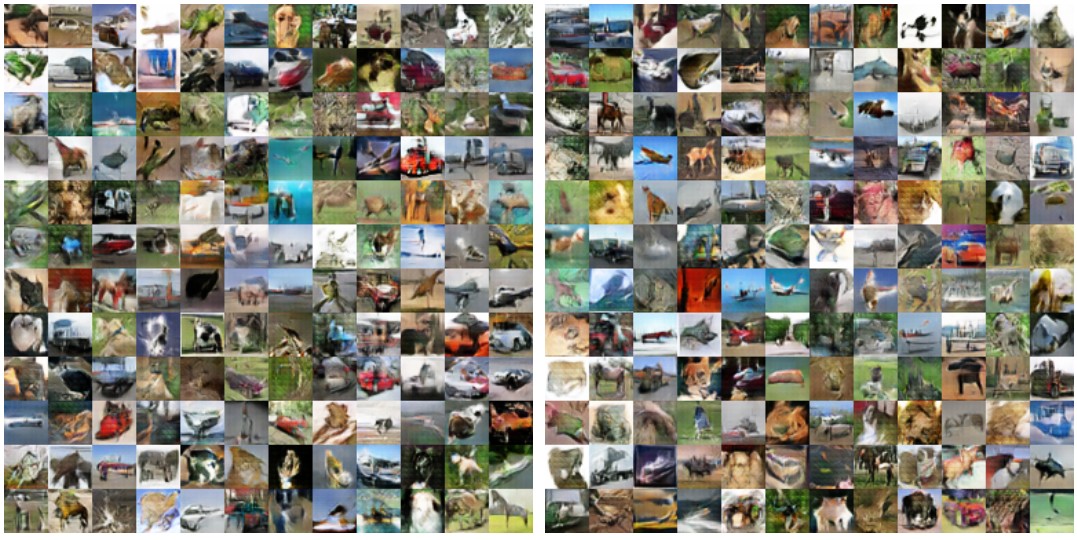

(a) Randomly sampled over all images

(b) Randomly sampled over images with top 50% densities

Figure 12: Generated images given by Joint-W on CIFAR.

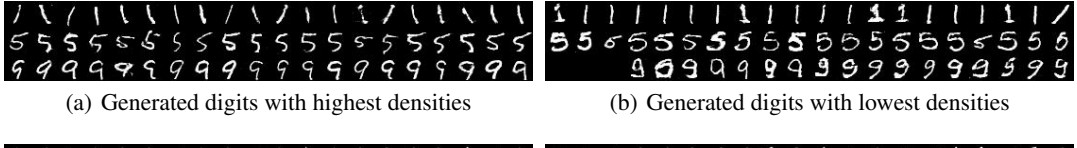

(a) Generated digits with highest densities          (b) Generated digits with lowest densities

(c) Real digits with highest densities          (d) Real digits with lowest densities

Figure 13: The generated digits (and real digits) with the highest densities and the lowest densities given by Joint-W.

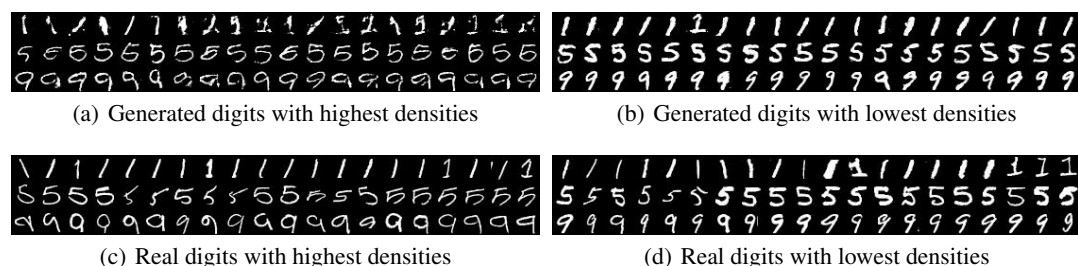

(a) Generated digits with highest densities          (b) Generated digits with lowest densities

(c) Real digits with highest densities          (d) Real digits with lowest densities

Figure 14: The generated digits (and real digits) with the highest densities and the lowest densities given by DGM.

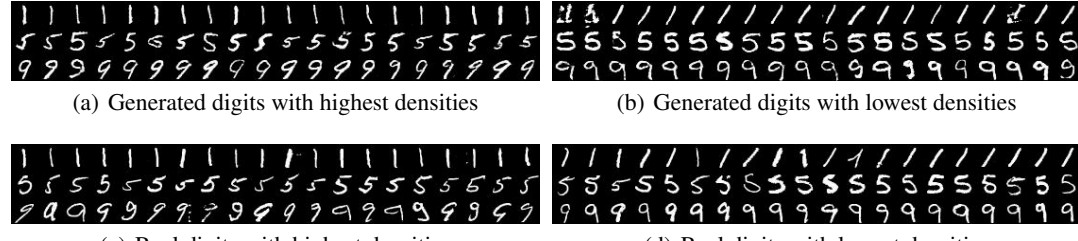

(a) Generated digits with highest densities          (b) Generated digits with lowest densities

(c) Real digits with highest densities          (d) Real digits with lowest densities

Figure 15: The generated digits (and real digits) with the highest densities and the lowest densities given by EGAN.