# OpenReview forum: "Bridging Explicit and Implicit Deep Generative Models via Neural Stein Estimators"
_NeurIPS.cc/2021/Conference — NeurIPS 2021 Poster_

### Official Review · Reviewer_p1LW · 2021-07-14

**Rating:** 6
**Confidence:** 3

**Summary:**

The authors propose to jointly train implicit (like GANs) and explicit (EBMs) density models by proposing a new regularization technique that connects the training of the impicit and explicit model via Stein bridges that is based on stein discrepancy. The authors show that such a learning objective with a Stein bridge helps to overcome the drawbacks encountered while training an implicit or explicit model individually. The authors provide sufficient theoretical details in support of the method under the Wasserstein distance and perform experiments on toy examples and image datasets.

**Limitations And Societal Impact:**

The authors have discussed limitations briefly towards the end but I believe the authors can include a more thorough discussion on the implications of their results in general, the limitations of theoretical analysis (restricted to only Wasserstein metric) and cost of training two models jointly and scalability to higher dimensions. The authors have alluded to some of the issues in the paper but I think detailed comments are warranted here.

**Main Review:**

**Pros:** The idea of using a combination of implicit and explicit models for joint training has been explored before in several papers recently (as the authors discuss in the detailed literature review section)  and is an interesting direction of research. Furthermore, the proposal to use mutual regularization of the two models to train such models is intriguing and novel. Additionally, the experiments presented in the paper provide promising results and improvements.

**Cons:**
- My major concern (which spans the previous works on jointly training implicit and explicit models as well) is the motivation behind the joint training of implicit and explicit model itself. It seems that this strategy introduces an expensive overhead of training two rather costly models to overcome the the drawbacks of the individual models. Furthermore, the solution itself does not directly address the drawbacks present in the individual models itself but indirectly makes things better for the models. While I liked the idea of incorporating the loss term between the implicit and the explicit model (D_3 in Table 1), I am still not clear what additional information this term provides during training apart from the data and the other two terms.

- Secondly, the authors mentioned in the paper that any metric can be easily used for training the models (apart from Wasserstein or Jenson-Shanon etc). However, if I am not mistaken, the theoretical analysis presented by the authors in Section 4 is based on the Wasserstein metric. I wonder if these results will hold for other metrics as well? What impact do the authors foresee of different metrics on their analysis?

- (*minor*): The authors train an implicit and explicit model together. Which of these models is finally used after training? Is it the case that the implicit model is used for sample generation and the explicit model for inference based tasks?

-  The methodology presented here comes across as rather too costly for training an energy based model specially given the fairly light-weight methods present for training EBMs. I believe the paper will benefit from having a comparison between the training using the proposed methodology and these already present methods for training EBMs.

- Finally, while the experiments comprehensively show that the proposed model achieves good results, I would have liked to see experiments on more complicated datasets than MNIST and CIFAR 10. This stems from my worry of the costly procedure of training two models jointly and how well this will scale to higher-dimensions and more complicated datasets.

**Time Spent Reviewing:**

~6

---

> ### Author Response · Authors · 2021-08-10
> **Response to Reviewer p1LW**
>
> Thank you for the positive comments and constructive feedback.
> Below we address the your concerns.
>
> **Q1**. Motivation and additional information the extra term provides during training.
>
> Thank you for this great question! We kindly remind that the motivation for combining the training of two classes of generative models not only lies in overcoming their respective limitations but also enables sample generation and density estimation in a unified framework. As discussed in line 30-37 of our paper, there are situations where one may need both samples and (unnormalized) densities at one time or the availability of both of them would be really helpful for some specific tasks. This serves as another motivation for combining both of the worlds in a new formulation.
>
> Indeed, there is no additional information brought by the third term since we do not use extra side information other than the observed samples as input. Yet, as shown in our theoretical analysis, the Stein bridging term acts as mutual regularization for two models which can help them to better identify the data distribution and generate more high-quality samples, compared with training a single model which has respective limitations (as discussed in Section 1 intuitively and Section 3 theoretically). Also, we show that the joint training approach can help to stabilize the training dynamics, which is another merit brought by the extra term.
>
> **Q2**. If the theoretical results hold for other metrics?
>
> A good question.
> Theorem 1 describes the impact of the Stein bridging on the critic. As a result, it can hold similarly if we replace the Wasserstein metric in Eqn. 3 with other metrics, as long as this metric admits a variational form that involves a critic (e.g., $f$-divergences). In this case, Eqn. 10 in Appendix will be changed into a slightly  different form, but the subsequent argument of the regularization effect of the Stein discrepancy on the critic should follow in a similar fashion.
>
> Theorem 2 relies on crucially the choice of metrics, and in particular, the Moreau-Yosida regularization comes from the proprieties of the Wasserstein metric. If we replace the Wasserstein metric in Eqn. 3 with other $f$-divergences, it may result in a different regularization behavior such as variance regularization.
>
> **Q3**. Which of these models is finally used after training?
>
> We use the explicit model for inference (density estimation) and the implicit model for sample generation. Your understanding is exactly right! This is indeed another merit for joint training two models, i.e., we can flexibly obtain estimated density and generated samples from two models simultaneously.
>
> **Q4**. Comparison with other methods for training EBM.
>
> This is a good point. Actually, our baseline DGM and DEM are two methods for training the EBM by contrastive divergence and using Stein discrepancy (i.e., what we use in our model), respectively. The difficulty for training EBM lies in how to handle the normalizing constant which is intractable in general cases. The Stein's method used in our framework is an effective way to detour the normalizing constant by considering the gradient of log-density, which avoids the heavy cost for estimating the denominator.
>
> **Q5**. Scalability of proposed method
>
> Actually, as illustrated in line 798, our model is two times slower than WGAN on average throughout the experiments. Our model can scale smoothly to larger datasets beyond MNIST/CIFAR-10 that we used. As a further validation, we also use LSUN dataset (a large-scale image dataset that is much more challenging than CIFAR-10) for our out-of-distribution detection task for testing the explicit model. Concretely, we treat CIFAR-10 as positive examples and LSUN images as negative examples. The model aims to distinguish the images from two datasets and is expected to give high density values for CIFAR-10 images and low density values for LSUN images. We compare our model with DEM, DGM and EGAN and obtain the results of Area Under of Curve (AUC): DEM 0.56, DGM 0.82, EGAN 0.56 and Joint-W 0.85. The results indicate that our model Joint-W can more accurately identify out-of-distribution samples from in-distribution ones, showing better empirical performance for density estimation in the high-dimensional inference tasks. We believe that these results can further strengthen our contribution.

---

### Official Review · Reviewer_cpPa · 2021-07-16

**Rating:** 5
**Confidence:** 4

**Summary:**

===== After Rebuttal =====

I have bumped up my score for the enhanced empirical results author(s) have provided in the discussion round. I remain conservative in my final recommendation because I am not fully convinced the gains are actually from the Stein energy penalty (i.e., some other regularization may also offer similar results) and the cost in modeling complexity may not be justified by the gains.

=====================

This paper proposes to improve generative modeling via joining the strength of unnormalized density estimation (based on Stein discrepancy) and Wasserstein GAN. Numerical experiments verify the proposed solution enjoys better convergence on both synthetic and real-world datasets. Some theories are developed to justify the use of training loss.

**Ethics Review Area:**

["Inadequate Data and Algorithm Evaluation"]

**Limitations And Societal Impact:**

- Missing citations in paragraph 2 in Introduction
- Line 34: "the generated samples from implicit models could augment the dataset and help to alleviate mode collapse". This is a false statement. This sounds more like one goal of generative modeling.
- Please explain why the Moreau-Yosida encourages the model to seek more modes. It is not obvious.
- Line 118, requiring the statistical distance to be computationally efficient seems unnecessary because the main bottleneck is the computation of Stein discrepancy, where gradient evaluations are involved.
- Line 140, "considers three discrepancies as a triangle", what does this line mean?
- Figure 4, all these competing solutions should work here, probably the author(s) should do a little bit of parameter tuning. Also, in Figure 5, the author(s) should compare the learned distribution from models in Figure 4.
- The statement in Theorem 1 seems to be different from the problem (3), which also minimizes for G.
- Experiments should include more complicated datasets. MNIST and Cifar10 do not adequately represent the complexity of real-world data, and that's what challenges Stein discrepancy based metrics.
- Examples of bad English:
"mutual compensation -> synergy"
"contaminated -> corrupted"
"power->strength"
"vivid->compelling"
"arguably->\"
"illuminate-> illustrates"
"oscillatory behavior->dynamics"
"centrality->solution"

**Main Review:**

This work seeks to combine energy modeling and implicit generative modeling to improve the learning of probabilistic models. This direction has been extensively explored by various researchers in the past two years, with varying degrees of success. While the insight that the proposed formulation indeed implements Moreau-Yosida regularization, my major criticism is that there are too many moving parts in this model and thus making it overly complicated. It feels like gluing the components together, and such bridging is not particularly meaningful. More discussion is needed to explain why this work is significant compared to prior arts.

Also, while the author(s) have claimed that the integrated solution overcomes the individual limits of each component, it can totally go to the other extreme, with the interacting components amplifying each other's deficiencies. No experiments are designed to verify this would not be the case (i.e., intentionally testing for the robustness). And there is an established consensus that working with Stein discrepancy is quite tricky. It is computationally intensive and scales poorly to data dimensionality and complexity (due to its kernel formulation and dependency on gradient computation). Since the author(s) are quite vague on the actual implementation of Stein discrepancy in this paper (for general Stein discrepancy, what is $\Gamma$, and for kernel Stein discrepancy, what kernel is used? ), I am unable to properly evaluate the effectiveness of the proposed Stein Bridging.

Originality. The techniques used in this work are very well known. The main novelty is to promote the consensus of model distribution and data distribution via minimizing the Stein discrepancy relative to the same energy model. I am not sure this is the right thing to do, as the consensus can be a bad one.

Quality & Clarity: Writing needs to be improved, many statements are not proper English. For clarity, details on Stein discrepancy, its implementation, and the reasoning why the proposed methods should function as expected are missing in the current presentation. I am not sure about the technical correctness of the theories.

Significance: This work is unlikely to impact the community in a significant way. The solution is overly complicated and the theories are neither appealing to theorists nor practitioners.

**Time Spent Reviewing:**

4

---

> ### Author Response · Authors · 2021-08-10
> **Response to Reviewer cpPa - Part 1**
>
> Thank you for your thorough review and thoughtful questions! Your feedback helps us to clarify our presentation, and we would like to take this opportunity to restate our contribution relative to the literature and  answer your questions below.
>
> **Response to Main Review**
>
> **Q1**. On the originality and significance of our work compared to prior arts.
>
> Our model introduces a new formulation that combines two classes of generative models via a Stein bridging term, as illustrated in the line 133-142 and a comparison provided in Table 1 where we compare with prior arts considering joint training of two generative models. Note that the advantage for combining two models not only lies in overcoming their respective limitations but also enables sample generation and density estimation in a unified framework. As discussed in line 30-37 of our paper, there are situations where one may need both samples and (unnormalized) densities at one time or the availability of both of them would be really helpful for some specific tasks. This serves as another motivation for combining both of the worlds in a new formulation.
>
> The key insight backed up with our theoretical analysis (in Section 3) is that the Stein bridging term serves as a regularization, i.e., kernel Sobolev norm penalization on the critic and smoothing the output of Stein critic via Moreau-Yosida regularization, which mutually reinforce the learning.
> We would like to point out that our main theoretical results (Theorems 1 and 2) are based on non-trivial optimization analysis, and to the best of our knowledge, these regularization effects brought by Stein bridging are new in the literature, and in particular, the Sobolev norm penalization deepens our understanding in using Stein discrepancy to facilitate the learning process.
>
> **Q2**. Details of implementation and experiments to verify the effectiveness of integrated solution.
>
> First, due to space limitation, some details for our method are deferred to the supplementary materials; the implementation details for Stein discrepancy in Appendix E.2 and E.3. For example, in line 784-788 of Appendix E.2, we present the kernel we used for kernelized Stein discrepancy, and in line 808-823 of Appendix E.3, we detailedly discuss how to implement general Stein discrepancy including the critic $\mathbf f$ and $\Gamma$.
> We kindly request you to refer to the Appendix for more details and let us know if further clarification is needed.
>
> In Section 4, we conduct experiments that compare our model with independently training either the explicit model or the implicit one. For example, in Fig. 4, we compare with DEM (which is an energy-based model trained with Stein discrepancy, i.e., only using the second term in Eqn. 3) and the results show that our model Joint-W manages to more accurately fit the ground-truth distribution. In Fig. 5, we compare with WGAN (which essentially only adopts the first term in Eqn. 3) and the results also show that Joint-W can generate more high-quality samples. The empirical results validate the superiority of our joint training method. Furthermore, in the MNIST/CIFAR experiments, we also compare Joint-W with WGAN-GP and DEM and our model yields significant improvement on both sample generation and density estimation tasks. Therefore, our experiments in Section 4 are carefully designed and demonstrate that our Stein Bridging can overcome the limitations of independently training one component and enable their mutual reinforcement.
>
>
> **Q3**. On the consensus of two generative models.
>
> We understand that it is a valid concern that consensus might be not ideal as it sounds like.
> However, based on our theoretical analysis and empirical findings, with proper joint training method the hybrid would be beneficial.
> The reason is that the additional divergence term does not change the global optimum where the distributions induced by both models match the data distribution. Notice that the two models are optimized in a cooperative manner with the same goal (i.e., minimizing the discrepancy from the data distribution) instead of an adversarial (zero-sum) game.
> Indeed, as shown in Section 3, the additional term brings up effective regularization effects for both models and helps to stabilize the training dynamics. Such results provide justification for pursuing a consensus of two in our formulation.
>
> Furthermore, empirically, our extensive experiments in Section 4 show that our model yields superior performance in various tasks compared with independently training a single generative model. Concretely, for density estimation, compared with the counterpart (i.e., the energy model) our model manages to 1) capture all the modes in the (unobserved) ground-truth distribution on two synthetic datasets in Fig. 4 where other competitors fail, and 2) provide much better performance for out-of-distribution detection in Table 3. For sample generation, compared with the counterpart (i.e., GAN/WGAN), our model 1) outputs samples that fit the (unobserved) ground-truth distribution well on two challenging synthetic datasets in Fig. 5 while other competitors fail to do so, and 2) yield superior inception score on CIFAR-10 as shown in Table 2. Besides, our further experiments in Fig. 6, Fig. 7 and Fig. 8 demonstrate that our model can help to stabilize the training dynamics and enhance the learning with contaminated/limited data.
> We believe that the above-mentioned results provide strong evidence for the effectiveness of enforcing the consensus of two models.
>
>
>
> **Reponses to Limitations.**
>
> **Q1, Q2 & Q9**. Missing citations and inproper statement/description.
>
> Thanks for pointing out this. We will add more citations and modify the suggested parts in the final version for more enjoyable presentation.
>
> **Q3**. How can Moreau-Yosida regularization encourage the model to seek more modes?
>
> This is indeed an important question. Our Theorem 2 shows that the Moreau-Yosida regularization can smooth the output of Stein critic, which helps it not to focus too much on some dominated modes and explore other potentially existing modes in the data; see see line 172-188 for details. Our example in Fig. 2 shows that such a regularization induced by our Stein bridge guides the energy model to identity the mode compared with the counterpart (not using the Stein bridge term) failing to do so. Furthermore, our experiments in Fig. 4 also verify this by comparing our model with DEM and other methods which fail to capture all the modes in data.
>
> Besides, we would like to point out that the Sobolev dual norm regularization in Theorem 1 ensures the critic $D$ not to change suddenly on high-density region of $\mathbb P_E$; see line 154-164 in our paper for more details. It prevents the Wasserstein critic from being too strong during training and thus alleviates the over-fitting of sample generator. Also we provide an illustration with an example in Fig. 3 (a) which shows the Sobolev norm regularization can reduce the gradient norms of the Wasserstein critic, which is in accordance with our result.
>
> These two mutual regularization effects provide a justification for our proposed formulation.
>
> **Q4**. Requiring the statistical divergence to be computationally efficient seems unnecessary.
>
> Thanks for raising this concern that helps to clarify our presentation.
> The reason why we mentioned computation efficiency is that in our context, the energy-based model has a normalizing constant which may introduce difficulty for optimization. For example, in MCMC, standard method like maximum likelihood estimation (i.e., KL divergence) needs to estimate the normalizing constant which could be highly time-consuming. The Stein discrepancy (what we use) can avoid calculating the denominator by handling the score function. As such, the "efficient computation" here means a statistical distance that can handle the normalizing constant well. We will clarify this point in the revised paper.
>
> **Q5**. What does "considers three discrepancies as a triangle" mean?
>
> We apologize for the vagueness. What we meant is that in Eqn. 3, the three terms incorporate the discrepancies between any pair of the three distributions $\mathbb P_{real}$, $\mathbb P_G$ and $\mathbb P_E$. We will modify the statement in the revision.
>
> **Q6**. Hyper-parameter tuning for methods in Fig. 4 and more comparison for methods in Fig. 5.
>
> Thank you for your suggestions.
> In fact, for all the baselines in Fig. 4, we did careful hyper-parameter tuning and we kindly refer to Appendix E.3 for detailed reports on the hyper-parameter settings.
> We think it is not fair to compare the learned distributions of models in Fig. 5 with those in Fig. 4. Note that in both figures, the input for each model are samples from a latent true distribution. Yet, the models in Fig. 4 are explicit models which can directly output estimated density but are hard for sample generation (except for DGM and our model Joint-JS/Joint-W), while the models in Fig. 5 are implicit models which are easy for sample generation but the density is not available. Therefore, in Fig. 4 we compare them on density estimation and in Fig. 5 we focus on the sample quality. For the baseline DGM that can simultaneously output estimated density and generated samples, we use it in both cases.
>
> **Q7**. The statement in Theorem 1 seems to be different from the problem (3).
>
> This is a good observation and is exactly the main point of Theorem 1, which solves out the minimization over $G$, provided that the generator is exhaustive. By solving out minimization over $\mathbb{P}_G$ explicitly we arrive at a regularization on the critic $D$.
> To the best of knowledge, the analysis based on infinite dimensional optimization and duality is non-trivial and the result is new in the literature.

---

> > ### Author Response · Authors · 2021-08-10
> > **Response to Reviewer cpPa - Part 2**
> >
> > **Q8**. Experiments should include more complicated datasets besides MNIST and CIFAR10.
> >
> > We agree that more complicated datasets would provide more evidence for supporting the effectiveness of our model. To this end, we also use LSUN dataset (a large-scale image dataset that is much more challenging than CIFAR-10) for our out-of-distribution detection task for testing the explicit model. Concretely, we treat CIFAR-10 as positive examples and LSUN images as negative examples. The model aims to distinguish the images from two datasets and is expected to give high density values for CIFAR-10 images and low density values for LSUN images. We compare our model with DEM, DGM and EGAN and obtain the results of Area Under of Curve (AUC): DEM 0.56, DGM 0.82, EGAN 0.56 and Joint-W 0.85. The results indicate that our model Joint-W can more accurately identify out-of-distribution samples from in-distribution ones, showing better empirical performance for density estimation. We believe that these results can further strengthen our contribution.
> >
> > We appreciate your thoughtful questions and hope our responses clarify our presentations and contributions.

---

> ### Author Response · Authors · 2021-08-22
> **Thank you for the time and we sincerely hope to receive your further comments**
>
> Dear Reviewer cpPa,
>
> We sincerely hope our posted response can help for addressing the lingering points of concerns and your re-assessment of our work. If you have any further comment and question, please let us know and we are glad to write a follow-up response. Below we provide a short-version summary for the motivation of our framework:
>
> - The extra Stein terms allows two classes of generative models to be jointly trained in a cooperative manner (Intuitively, the explicit one plays as a sample evaluator and the implicit one can augment the observed samples).
> - The unfied framework enables (unnormalized) density estimation and sample generation at one time, which is beneficial for various tasks that require both of them.
> - We theoretically justify the approach by showing that the joint learning enables mutual compensation between two models via regularization effects.
>
> Thank you very much for the time reading our comments.

---

> > ### Comment · Reviewer_cpPa · 2021-08-26
> > **Thanks for the rebuttal**
> >
> > Thank you for the very detailed response, I appreciate it. I confirm I have carefully read the author(s)' rebuttal along with other reviewers' comments. I am willing to adjust my score to 5 to reflect the changes made and promised. The main justification for the re-evaluation of this work is that the author(s) have provided additional results on more challenging real-world datasets, so it does have some practical merits. Note that my concerns on the limitations of kernel-based Stein discrepancy metrics still remain. After all, it is still a computationally demanding metric, and I wonder if simpler objectives like [Alain, et al. (2014)]  can have a similar result. This will help to clarify whether it is the idea of joint training of energy & generative models or the gain is only due to the Stein regularization.
> >
> > [Alain, et al. (2014)] What regularized auto-encoders learn from the data-generating distribution. JMLR

---

> > > ### Author Response · Authors · 2021-08-28
> > > **Thank you for the feedbacks and a follow-up response**
> > >
> > > Thanks for your valuable feedbacks and comments. We provide more illustrations to address your concern.
> > >
> > > We would like to first point out that in the context of our formulation, the Stein discrepancy is an ideal choice for the bridging term to measure the discrepancy between the implicit generative model and the explicit energy model. Recall that the implicit model is flexible for sampling (but hard for density estimation) while the energy model can produce unnormalized densities (but hard for generating sample and its normalizing constant is often intractable for computation). As such, the Stein discrepancy is suitable since it only requires samples from the implicit distribution and the gradient of log-density from the explicit distribution.
> > >
> > > We indeed follow the work of [Alain, et al. (2014)] to perform the following experiment that can demonstrate the effectiveness of Stein regularization in our approach. Specifically, we consider the baseline WGAN+VA [Tao et al. (2019)] whose objective has a WGAN term plus a surrogate likelihood term estimated based on the formulation proposed in [Alain, et al. (2014)]. We compare this baseline in our synthetic datasets (Table 6), image datasets (Table 2) and stability test (Fig. 7). The results show that our approach yields consistent performance gain for sample generation. We believe these results provide some evidence for the superiority of Stein regularization.
> > >
> > > We would also like to add a remark on the issue of the high computational cost of kernelized Stein discrepancy. First, we adopt kernelized Stein discrepancy only in our experiments on synthetic datasets, since this is a 2-dim task and the computation is not heavy. It seems that using KSD is not time-consuming and meanwhile can produce effective performance in such a low-dimensional task. Second, in our experiments on image datasets, we adopt the neural-network-based Stein discrepancy (with Stein critic), and we suggest using $d'=1$, which we found works as well as using larger $d'$ and can reduce the computational cost. In such a case, our approach is two times slower than WGAN-GP on CIFAR-10 (i.e., Table 2), which is an acceptable budget.
> > >
> > > Hope these concrete results can help to address your concerns on the adoption of Stein discrepancy.
> > >
> > > [Tao et al. (2019)] Variational Annealing of GANs: A Langevin Perspective, in ICML'19.

---

> > > > ### Author Response · Authors · 2021-09-01
> > > > **Thank you for the time and we would like to see if there is any further concern**
> > > >
> > > > Dear Reviewer cpPa,
> > > >
> > > > Since it is approaching the end of the discussion period, we would like to kindly ask if our previous response clarifies your concerns on KSD and Stein regularization as compared to [Alain, et al. (2014)], and if there are any further questions that we could answer to facilitate the review process. Thanks a lot for your help!

---

> > > > > ### Comment · Reviewer_cpPa · 2021-09-02
> > > > > **Thanks for the clarification**
> > > > >
> > > > > Sorry for the delay in responding, being overwhelmed by work lately. I have read your follow-up clarification and they do make sense. That said, I wish to keep my current score, which already stretched the limit of my comfort zone. As pointed out in my original review, while there are some theoretical arguments to favor Stein bridging, there are many moving parts in this composed loss function. My feeling is that energy modeling does help implicit generative modeling, but we need to integrate the two in a more organic manner. The current study still takes a more traditional regularization perspective so it hurts the novelty part in my overall evaluation. I acknowledge that this will be challenging, and I personally do not have much clue on how to bridge energy modeling and generative modeling without an explicit penalty. Perhaps sharing model parameters would be a sensible alternative.

---

> > > > > > ### Author Response · Authors · 2021-09-02
> > > > > > **Thank you for the feedbacks and shared insights**
> > > > > >
> > > > > > Thanks for your comments. We agree that it is a non-trivial problem for combining the training of two classes of generative models in an elegant way and indeed it is an open problem with significance. We believe our paper opens a possibility and explores a new perspective by using Stein bridging, which is backed up with our theoretical analysis and extensive experiments. The accompanying theory reveals its mutual regularization effects between energy model and the implicit generative model, i.e., 1) the energy model indeed helps the implicit model by penalizing large kernel Sobolev norm of the critic, and 2) the implicit model indeed helps the energy model by smoothing the function of Stein discrepancy through Moreau-Yosida regularization. These results play as a grounded justification for our method and shed new insights on the rationale of Stein bridging for generative modeling. Furthermore, our empirical results demonstrate that our proposed approach indeed yield mutual reinforcement between two models. The energy model can provide better density estimation performance while the implicit model can generate higher-quality samples.
> > > > > >
> > > > > > To sum up, our work explores a new method which is shown to be theoretically grounded and yields promising empirical results. We believe these contributions could bring up significance and impacts for frontiers of generative modeling

---

### Official Review · Reviewer_c3DX · 2021-07-16

**Rating:** 6
**Confidence:** 3

**Summary:**

Explicit and implicit model training have their own advantages and limitations. The author proposes a joint training framework, which aims to combine the benefits from both worlds. Specifically, the author proposes a training framework called **Stein bridging** that allow joint learning of explicit and implicit models.  Particularly, the explicit model considered in this paper is the energy-based model (EBM) and the implicit model is the GAN-like generator. The implicit model is trained based on GAN framework, the explicit model is learned via Stein discrepancy, and the bridge between them is also the Stein discrepancy between the generated samples and the explicit model.

Theoretically, the author proves that the Stein bridging can (1) act as a dual Sobolev norm regularization for the discriminator that helps the GAN training; (2) act as a Moreau-Yoshida regularization for the Stein discrepancy to smooth the energy function; (3) also help to stabilize the training dynamics.

Empirically, the author conducts extensive experiments to confirm the effectiveness of the proposed bridging, in terms of generating quality and OOD detection.

**Limitations And Societal Impact:**

This paper is a methodology paper. So there shouldn't be any negative societal impacts.

**Main Review:**

### Originality
The main idea of briding is simple and straightforward. However, the theoretical analysis is the main contribution of the paper and seems to be non-trivial. It provides some insights into the regularization effect of the Stein bridging. The author also addresses the differences between this work and the previous ones in the appendix. So in summary, I think this work is interesting and novel.

### Quality and Clarity
The main idea is easy to follow and clearly written. Some of the simple toy examples (from Theorem 1 and Theorem 2) help to visualize the effect of the regularization. However, I have several concerns regarding the Stein discrepancy and proof. I think there is still a large room for improvements of the quality. I will elaborate on them in the question section.

### Significance
This proposed framework seems to have many desirable advantages in terms of implicit model training stability and explicit model training quality. In terms of theory, the author shed some light on the effect of joint learning of these two types of models. So it should have some impacts in this community.

### Questions
1. The first question regards the usage of Stein discrepancy. Using Stein discrepancy to train models can be quite problematic. For example, the Stein discrepancy can be highly non-convex and does not agrees with commonly used divergence (e.g. KL divergence). In other words, when use Stein discrepancy as the training objective (in you case, the learning of explicit model and Stein bridging), a lower discrepancy value does not necessarily mean a good model (the two densities $P_E$ and $P_G$ can be arbitrarily far away in KL divergence sense) unless the global optimum is achieved [1]. This is particularly problematic for gradient-based optimization. So does this pathology of Stein discrepancy affect the Stein bridging? Or the Stein bridging only act as regularization and has no direct impact to model learning.
2. One quick check for question 1 could be replacing the components in Two-Circle example by Student-t distributions. The EBM is initialized as a mixture of Student-t distribution with the same scale and degree-of-freedom but different means (need to be far-away from the ground truth).  I wonder about the behaviours of the explicit and implicit models, and its training dynamics.
3. For the definition of Stein discrepancy, when $d'\neq d$, is the corresponding Stein discrepancy a valid one? Does the max-pooing matrix norm guarantee a valid discrepancy?
4. I am a bit confused on why Sobolev dual norm penalizes the large variation of critic. Maybe consider adding some explanation on the intuition behind this norm?
5. Maybe consider adding some intuitive explanations on Moreau-Yoshida regularization?
6. What is $p_r$ in line 615, is it $P_{real}$?
7. For KSD, the form mentioned below line 619 is different from the one you introduced in the main text. I understand this comes from integration by parts, but maybe explicitly mention this before line 619? Also, should it be $\nabla_{\pmb{x}}\log d\mathbb{P}_E/d\mathbb{P}$?
8. In line 629, why it is linear w.r.t. $h$? $h$ appears inside the $\log$.
9. I am confused about the derivation below the line 634. Why you can introduce the auxiliary variable $r$ and treat it as a decoupled parameter? $r$ should also depend on $h$ as well.
10. For line 653, why $P$ is unconstrained? I thought that $P$ should be absolutely continuous to $P_E$, but $\gamma$ only cares about the agreement of the marginal $P$ and $P_{real}$. So it is possible that $P$ is not absolutely continuous to $P_E$.
11. Why it is equivalent to the set of point masses in line 660?


### Reference
[1] Barp, Alessandro, et al. "Minimum stein discrepancy estimators." arXiv preprint arXiv:1906.08283 (2019)

---
### Upates after rebuttal
I appreciated the response. After some discussions with the author, they managed to address my concerns. Therefore, I will raise my score accordingly. However, I still hope more intuitions and explanations about the kernel Sobolev and Moreau-Yoshida regularization can be added in the revised version.


**Time Spent Reviewing:**

6

---

> ### Author Response · Authors · 2021-08-10
> **Response to Reviewer c3DX**
>
> We appreciate very much for your careful and thorough review as well as your constructive feedback! They greatly help to improve the presentation of our paper.
> Below we respond to each of your concerns.
>
> **Q1**. Does the pathology of Stein discrepancy affect the Stein bridging?
>
> This is a good question. First, we agree that having a probability distance measure with nice differentiability with respect to parameters is good for gradient-based optimization, and that's one of the main motivation of incorporating Wasserstein metric in our training objective. That said, as pointed out in the WGAN paper, many divergence measures do not enjoy good differentiability properties for neural-net generators, but this alone does not necessarily prevent researchers from using divergence measures in practice. Therefore, we believe Stein discrepancy still has its merits, for example, it does not require the knowledge of normalizing constant of a distribution.
>
> Second,
> in our experiments on two synthetic datasets, we do observe that directly training an energy model (i.e., the DEM in Fig. 4), using the Stein discrepancy between model's (unnormalized) distribution and the true samples, cannot provide desirable performance. As shown in Fig. 4, DEM only captures partial modes of ground-truth distributions, i.e., stuck in local optimum. By contrast, our models Joint-JS and Joint-W manage to identify all the modes and accurately fit the ground-truth. The results validate that Stein Bridging enhances the learned densities of energy model and helps to resolve the pathology of Stein discrepancy used for explicit model through optimization.
> Therefore, to your question, Stein Bridging acts more like regularization.
>
>
> **Q2**. More experiments on Two-Cricle dataset with Student-t distribution
>
> Thank you for your kind suggestions, as it is always good to include more evidence for testing our framework. We conduct a new experiment using Student-t distribution (as suggested by the reviewer) to replace the Gaussian in our Two-Cricle dataset. We observe similar behaviors in such a case, i.e., our model manage to identify the 24 modes in data and the DEM fails with only 7 modes captured. We believe that the new results can further back up our insight.
>
> **Q3**. Is the Stein discrepancy a valid one when $d'\neq d$?
>
> The answer is true. Note that the return of the Stein operation is a matrix. Concretely we have $\mathcal A_q[\mathbf f(\mathbf x)] \in \mathbb R^{d'\times d}$ where $\mathbf x\in \mathbb R^d$ and $\mathbf f: \mathbb R^d\rightarrow \mathbb R^{d'}$. Therefore, we need an operation $\Gamma$ to transform the matrix to a scalar. The Stein discrepancy is still a valid one once the property also holds that $\sup_{\mathbf f\in\mathcal F} \Gamma \left( \{\mathbb E_{\mathbf x\sim q}[\mathcal A_p[\mathbf f(\mathbf x)]]\} \right )=0$ if and only if $p$ and $q$ is a same distribution. For example, we can consider $\Gamma$ as matrix norm. Yet, using matrix norm would bring up difficulty for stochastic optimization since it cannot be altered with the expectation, which hinders mini-batch gradient descent. Therefore, we specifiy it as max pooling in practice. Such a choice keeps the global optimum unchanged and we found it works smoothly throughout our experiments.
>
> **Q4**. Intuitions about Sobolev dual norm penalization.
>
> The Sobolev dual norm regularization ensures $D$ not to change suddenly on high-density region of $\mathbb P_E$, and this is what we meant by penalizing large variation of the Wasserstein critic; see line 154-164 in our paper for more details. It prevents the Wasserstein critic from being too strong during training and thus alleviates the over-fitting of sample generator. Also we provide an illustration with an example in Fig. 3 (a) which shows the Sobolev norm regularization can reduce the gradient norms of the Wasserstein critic, which is in accordance with our result.
>
> **Q5**. Intuitions about Moreau-Yosida regularization.
>
> The Moreau-Yosida regularization of the Stein Bridging plays as a Lipschitz regularization on the output of the Stein operator, because the output is always Lipschitz with constant controlled by $\lambda_2$; see more details in line 172-183. By smoothing the output of Stein critic, it encourages the energy model to seek more modes in data instead of focusing on some dominated modes, thus alleviating mode-collapse issue. More illustration is provided in Fig. 2 where we show how the Stein bridge helps to smooth the Stein critic and encourage the model to identify the mode in data.
>
> **Q6**. What is $p_r$ in line 615?
>
> Indeed, $\mathbb P_r$ should be $\mathbb P_{real}$ in line 615. Thanks for spotting this typo.
>
> **Q7**. The inconsistent form of KSD below line 619.
>
> Thank you for your suggestion! Yes, indeed
> the form below line 619 follows from an equivalent expression of KSD (see, for example, Definition 3.2 in [27]). We will explicitly mention this in the revision. It can be either $\nabla_{x}\log d\mathbb P_E/d\mathbb P$ or $\nabla_{x}\log d\mathbb P/d\mathbb P_E$ because they differ by a negative sign and are equivalent after taking the quadratic form.
>
> **Q8**. Why is it linear w.r.t. $h$ in line 629?
>
> Nice catch! Please allow us to modify the argument in line 622-630 as below. Introducing a change of variable $H(\mathbf{x})=\log d\mathbb P/d\mathbb P_E$, then problem (10) becomes
>
> $
>     \min_{E,h}\max_D \left\\{ \mathbb E_{\mathbb P_E}[e^H D] - \mathbb E_{\mathbb P_{real}}[D] + \lambda_1 \mathcal{S}(\mathbb P_{real},\mathbb P_E) + \lambda_2 \mathbb E_{\mathbf{x},\mathbf{x}'\sim\mathbb P}[\nabla_x H(\mathbf{x})^\top k(\mathbf{x},\mathbf{x}')\nabla_x H(\mathbf{x}')] \right\\}.
>   $
>
>   Fixing $E$, the objective function is linear in $D$ and convex in $H$ due to the convexity of the exponential function, the linearity of expectation operator and differential operator, and the positive definiteness of $k$.
>   Then using a similar argument in line 625-630 we can justify the exchange of $\min_H$ and $\max_D$. Finally, noticing that $H=\log(1+h)$ where $h$ is defined in line 623, we can proceed with the proof starting from line 631.
>
> **Q9**. Elaboration for auxiliary variable $r$.
>
> By introducing an auxiliary variable $r$, the objective becomes decoupled but the two terms are linked through the constraint. The first equality below line 634 holds because the minimization over $r$ forces $r^2 = \mathbb{E}_{\mathbf{x},\mathbf{x}'\sim\mathbb{P}_E}[\nabla_x h(\mathbf{x})^\top k(\mathbf{x},\mathbf{x}') \nabla_x h(\mathbf{x}')]$ at optimality.
>
> **Q10**. Why $P$ is constrained in line 653?
>
> Observe that by definition,
> $\mathbb E_{\mathbb P}  [\mathcal{A}_{\mathbb P_E}\mathbf{f}(\mathbf{y})]$ equals
>
> $\mathbb E_{\mathbb P} [\nabla_y\log d\mathbb P_E/d\mathbb P(\mathbf{y})\mathbf{f}(\mathbf{y})^\top +\nabla_y \mathbf{f}(\mathbf{y})]$, which is infinite if $\mathbb P$ is not absolutely continuous in $\mathbb P_E$. In the meantime, (12) is a minimization problem over $\mathbb P$, hence those $\mathbb P$'s that are not absolutely continuous in $\mathbb P_E$ are automatically ruled out.
>
>
> **Q11**. Why it is equivalent to the set of point masses in line 660?
>
> Line 660 holds because the inner minimization over $\gamma_{\mathbf{x}}$ in the second line below 656 can be attained at a Dirac mass concentrated on the minimizer $\arg\min_{\mathbf{y}}\{\lambda_2\mathcal{A}_{\mathbb{P}_E}\mathbf{f}(\mathbf{y})+ ||\mathbf{x}-\mathbf{y}||\}$, provided that the minimizer exists; otherwise we can use an approximation argument to show it suffices to only consider point masses.
>
> Thanks again for your great observations and thoughtful questions!
> Please let us know if you have any further questions and comments.

---

> > ### Comment · Reviewer_c3DX · 2021-08-23
> > **Student-t experiments**
> >
> > I appreciate the detailed rebuttals. It addresses most of my concerns. However, I still have the following two remaining questions.
> >
> > 1. I am surprised about the student-t experiment, especially the DEM trained with Stein discrepancy. According to Barp., it should find any mode at all. Can you provide the experiment details and any guesses on the disagreement between yours and Barp.?
> >
> > 2. For Q3, maybe I missed your point. My concern is there is no proof that this max-pooling $\pmb{\Gamma}$ satisfies the property you mentioned. For example, in the original KSD paper [Liu., et al, 2016], they firstly choose $\pmb{\Gamma}$ to be the trace operator. Then derive the KSD and show that KSD is a valid discrepancy. In Stein neural sampler [Hu., et al, 2018], the form of Stein discrepancy is also based on trace operator, and then they show it is a valid discrepancy. Thus, I think after defining the operator $\pmb{\Gamma}$, you need to show the resulting discrepancy is valid (I understand when $d=d'$, your operator is stronger than the trace operator, but what about $d\neq d'$?).

---

> > > ### Author Response · Authors · 2021-08-23
> > > **Response to the follow-up questions**
> > >
> > > Thank you for the follow-up feedback. Below we provide further details regarding your questions.
> > >
> > > **Q1**: Details for Student-t experiments and discussions
> > >
> > > Thank you for raising up this thoughtful question. If we interpret your question correctly, you asked about the different performance of the energy models between Barp's work and ours.
> > > Our understanding is that our experiment setups are quite different, which explains why we see somewhat different trends and phenomenons.
> > > Below we compare the setup, implementation details and results in Barp' paper and ours.
> > >
> > > **Barp's numerical experiments in Section 4.2**. They assume a student-t distribution (which we call ground-truth distribution) for generating a batch of observed samples. Then they consider another student-t distribution (which we call model distribution) with learnable location/scale parameters. The KSD with IMQ kernel is adopted to measure the discrepancy between observed samples and model distribution, and then SGD (or Riemannian SGD) algorithm is used for minimizing the KSD loss. After that, the authors compare the learned location/scale parameters of model distribution and the counterpart of ground-truth distribution. Their observation is that minimization of KSD loss with SGD fails to find the ideal solution.
> > >
> > > **Ours**. In our case, we assume a mixture of 24 student-t distributions (as ground-truth distribution) for generating 2000 observed samples. Then we leverage a deep energy model as model distribution. The DEM takes a sample as input and outputs a scalar probability value. Following the reviewer's suggestion, the DEM is implemented in this way: it first maps an input sample to a 24-dim vector (each entry corresponds to the location estimation for one student-t distribution) using a three-layer neural network and then weighted sum the 24 student-t probabilities to get the output scalar. Then we use KSD + RMF kernel to measure the discrepancy between observed samples and model distribution and harness RMSProp optimizer for training. Finally, we compare the densities output by DEM and the ground-truth one. Our observation is that the high-density regions given by DEM only match 7 out of 24 student-t modes of the ground-truth distribution.
> > >
> > > Therefore, there are several notable differences between two cases, especially the parametric model distribution, training algorithm and evaluation. In Barp's experiment, they aim to learn a parametric form of distribution (with two distributional parameters) while in our case we need to train a neural network energy model (with many parameters). Also, we use RMSProp for training while they use SGD or Riemannian SGD, which may also lead to distinct behaviors in optimization. Furthermore, in Barp's work they can directly calculate the accuracy of the estimated parameter, while in our case the DEM is a black-box and we need to compare the output densities. Therefore, we believe the above distinctions explain the difference between our result and Barp's.
> > >
> > > **Q2**: Stein discrepancy for $d \neq d'$
> > >
> > > Thanks for pointing out this and we apologize for our imprecise description in previous response. The $\Gamma(\cdot)$ is specified as $maxpooling(abs(\cdot))$ in our experiments on CIFAR-10 instead of a pure maxpooling. We next show that this can guarantee a valid discrepancy. On one hand, from Stein's identity, if $f$ belongs to the Stein class then $\mathbb E_p[\mathcal A_p \mathbf f]$ is a $d'\times d$ zero matrix, and thus taking the maxpooling yields a zero scalar value. Therefore, if $p=q$, we have $maxpooling(|\mathbb E_q[\mathcal A_p \mathbf f]|)=0$.
> > > On the other hand, if the maxpooling of an element-wise non-negative matrix is zero, then this matrix is a $d'\times d$ zero matrix.
> > > When $\mathcal F$ is sufficiently large, $\sup_{\mathbf f\in\mathcal F} |\mathbb E_q[\mathcal A_p \mathbf f]|$ being a zero matrix implies $p=q$. Therefore, if $\sup_{\mathbf f\in\mathcal F} maxpooling(|\mathbb E_q[\mathcal A_p \mathbf f]|)=0$, due to the fact $\sup_{\mathbf f\in\mathcal F} maxpooling(|\mathbb E_q[\mathcal A_p \mathbf f]|)=maxpooling(\sup_{\mathbf f\in\mathcal F} |\mathbb E_q[\mathcal{A}_p \mathbf f]|)$ where the supremum is interpreted element-wise, then we have $p=q$.
> > >
> > > Hence, we have shown that $\sup_{\mathbf f\in\mathcal F} maxpooling(|\mathbb E_q[\mathcal{A}_p \mathbf f]|)=0$ if and only if $p=q$, as long as $\mathcal F$ is large enough (we specify $\mathbf f$ as a neural network to get enough capacity).
> > > We remark that Hu et al. (2018) also requires a large $\mathcal{F}$ when using trace operator. Practically we found it works smoothly. Therefore, our used Stein discrepancy for $d \neq d'$ is a valid discrepancy.
> > >
> > > We hope the response can address all of your concerns. If you have any further question, please let us know.

---

> > > > ### Comment · Reviewer_c3DX · 2021-08-23
> > > > **Validity of Stein discrepancy**
> > > >
> > > > I understand when $p=q$, the $d\times d'$ matrix is $0$. How do you show when it is a zero matrix, $p=q$? Any references?

---

> > > > > ### Author Response · Authors · 2021-08-24
> > > > > **Further response**
> > > > >
> > > > > From our understanding, your concern lies in the argument in our above reasoning: When $\mathcal F$ is sufficiently large, $\sup_{\mathbf f\in\mathcal F} |\mathbb E_q[\mathcal{A}_p \mathbf f]|$ being a zero matrix implies $p=q$.
> > > > > Indeed, it is a bit vague by saying '$\mathcal F$ is sufficiently large', although similar statements are also made by [1] (see Eqn. 2) and [2] (see Eqn. 2.2) when defining the Stein discrepancy with Stein class.
> > > > > This argument acts more like a 'condition' for which one can construct proper $\mathcal F$ to meet the requirement, such as RKHS or neural nets.
> > > > > We believe the desired implication depends on the distributions $p,q$ and the choice of $\mathcal F$, and hence there is no universal result that works for arbitrary distributions or $\mathcal F$ (e.g., in [1, 3] the authors discuss various ways for construction of $\mathcal F$ w.r.t. different distribution forms).
> > > > >
> > > > > Below let us show a simple illustration on how the implication can be achieved in our case by choosing a sufficiently large $\mathcal F$. Consider, for example, two positive and smooth densities $p,q$ on a compact set $\mathcal X\subset\mathbb R^d$ with a smooth boundary $\partial\mathcal X$. Then a function $f$ is in the Stein class of $p$ if $f(x)p(x)=0$ for every $x$ on the boundary $\partial\mathcal X$ (see, for example, the Remark on page 3 of [4]), and by Lemma 2.3 in the same reference, we have
> > > > > $
> > > > >   \mathbb E_p [\mathcal A_q \mathbf f] = \mathbb E_p [(\nabla \log q - \nabla \log p) \mathbf f^\top].
> > > > > $
> > > > >
> > > > > Consider $\mathcal F=\\{\mathbf f: \mathcal X\to \mathbb R^{d'}: |\mathbf f_j(x)| \le 1, \mathbf f_j(x)=0,\forall x\in\partial\mathcal X, 1\le j \le d' \\}$.
> > > > > If $\sup_{\mathbf f \in \mathcal F} maxpooling(|\mathbb E_p [\mathcal A_q \mathbf f]|)=0$, then
> > > > >
> > > > > $\sup_{\mathbf f \in \mathcal F} (|\mathbb E_p [\mathcal A_q \mathbf f]|)_{ij}=0$, for every $i=1,\ldots,d$, $j=1,\ldots, d'$,
> > > > >
> > > > >  where the subscript $\cdot_{ij}$ denotes the $i$-th row and $j$-th column of a matrix.
> > > > > Observe that for every $i=1,\ldots,d$, $j=1,\ldots, d'$,
> > > > >
> > > > > $
> > > > >   \sup_{\mathbf f \in \mathcal F} (|\mathbb E_p [\mathcal A_q \mathbf f]|)_{ij} = \mathbb E_p [|\nabla_i \log q - \nabla_i \log p|],
> > > > > $
> > > > >
> > > > > where $\nabla_i$ denotes the gradient with respect to the $i$-th coordinate.
> > > > > Hence it is zero if and only if $p=q$ on $\mathcal X$.
> > > > >
> > > > > [1] Anastasiou et al., Stein’s Method Meets Statistics: A Review of Some Recent Developments.
> > > > >
> > > > > [2] Hu et al., STEIN NEURAL SAMPLER.
> > > > >
> > > > > [3] Gorham et al., Measuring Sample Quality with Stein’s Method.
> > > > >
> > > > > [4] Liu et al., A Kernelized Stein Discrepancy for Goodness-of-fit Tests.

---

> > > > > > ### Comment · Reviewer_c3DX · 2021-08-24
> > > > > > **Validity of Stein discrepancy**
> > > > > >
> > > > > > Thank you for your response. I do not understand your last argument.  Why the supremum over $ \sup_{f\in\mathcal F}(|\mathbb E_p[\mathcal A_q f|)$ equals $\mathbb{E}_{p}[|\nabla_i\log q-\nabla_i\log p|]$ for all $i,j$? Maybe I misunderstood something. For example, let's denote $\Delta_i \pmb{s}=\nabla_i\log q - \nabla_i \log p$. So we can write $\mathbb E_p[\Delta \pmb{s}\pmb{f}^\top]$ as $\mathbb E_p[[\Delta_1 \pmb s,\ldots,\Delta_D \pmb s]^T\pmb{f}]$. So for $i=1$, to achieve your argument, the $\pmb f$ has to be $\Delta_1 \pmb s$. If so, for $i=2$, your argument won't hold. In summary, my question is what is the form of $\pmb{f}$ to achieve your argument?

---

> > > > > > > ### Author Response · Authors · 2021-08-24
> > > > > > > **Optimal $\mathbf f$**
> > > > > > >
> > > > > > > Sorry for the ambiguity. Please allow us to add more details.
> > > > > > >
> > > > > > > First, observe that for every $i$, $j$, we have a scalar equation $$(\mathbb E_p [\mathcal A_q \mathbf f])_{ij} = \mathbb E_p[(\nabla_i \log q - \nabla_i \log p) \mathbf f_j].$$
> > > > > > >
> > > > > > > Thus $\\sup_{\\mathbf f \\in \\mathcal F} |(\\mathbb E_p [\\mathcal A_q \\mathbf f])_{ij}| $ attains its supremum at $\mathbf f_j(x) = sgn(\nabla_i \log q(x) - \nabla_i \log p(x))$ or $\mathbf f_j(x) =- sgn(\nabla_i \log q(x) - \nabla_i \log p(x))$, where $sgn$ denotes the sign function, and the supremum equals the expectation (w.r.t. $p$) of the absolute difference between $\nabla_i \log q - \nabla_i \log p$.
> > > > > > >
> > > > > > > Also note that this is only true for a fixed pair of $(i,j)$, and we did not try to solve $\\sup_{\\mathbf f \\in \\mathcal F} maxpooling(|\\mathbb E_p [\\mathcal A_q \\mathbf f]|)$ in general.
> > > > > > >
> > > > > > > Second, if it holds that $$ \\sup_{\\mathbf f \\in \\mathcal F} (|\\mathbb E_p [\\mathcal A_q \\mathbf f]|)_{ij}=0,$$
> > > > > > >
> > > > > > > then $\\sup_{\\mathbf f \\in \\mathcal F} (\\mathbb E_p [\\mathcal A_q \\mathbf f])_{ij}=0$. From the first step we have
> > > > > > >
> > > > > > > $$\\mathbb E_p [|\nabla_i \log p - \nabla_i \log q|]=0,$$
> > > > > > >
> > > > > > > which implies $\nabla_i \log q = \nabla_i \log p$ on $\mathcal{X}$. The above analysis applies to every $i$, therefore we obtain that $\nabla \log q = \nabla \log p$. Since $p,q$ are densities, we conclude that $p=q$.
> > > > > > >
> > > > > > > We appreciate your time, and please let us know if there is any other concern.

---

> > > > > > > > ### Comment · Reviewer_c3DX · 2021-08-24
> > > > > > > > **Validity of Stein discrepancy**
> > > > > > > >
> > > > > > > > Hi, I am still confusing about the argument. When $d\neq d'$ (let's assume $d'<d$), then how can you make $f_j=sgn(\nabla_i \log q-\nabla_i \log p)$ since the size of $i$ and $j$ does not match.

---

> > > > > > > > > ### Author Response · Authors · 2021-08-24
> > > > > > > > > **Clarification**
> > > > > > > > >
> > > > > > > > > Here $i,j$ are fixed. $\mathbf f_j:\mathcal X \to \mathbb [-1,1]$ is a real-valued function, and $\nabla_i \log q - \nabla_i \log p$ is also a real-valued function. Any vector-valued function $\mathbf f$ whose $j$-th component equals $\pm sgn(\nabla_i \log q - \nabla_i \log p)$ would be an optimal solution to the problem $\\sup_{\\mathbf f \\in \\mathcal F} (|\\mathbb E_p [\\mathcal A_q \\mathbf f]|)_{ij}$ .
> > > > > > > > >
> > > > > > > > > As we noted in the previous reply, we did not try to solve $\\sup_{\\mathbf f \\in \\mathcal F} maxpooling(|\\mathbb E_p [\\mathcal A_q \\mathbf f]|)$ in general, but only solve for $\\sup_{\\mathbf f \\in \\mathcal F} (|\\mathbb E_p [\\mathcal A_q \\mathbf f])_{ij}$ for a fixed pair of $(i,j)$. In fact, all we need to know is that
> > > > > > > > >
> > > > > > > > > $$\\sup_{\\mathbf f \\in \\mathcal F} (|\\mathbb E_p [\\mathcal A_q \\mathbf f])_{ij}=0$$
> > > > > > > > >
> > > > > > > > > forces $\nabla_i \log p = \nabla_i \log q$.
> > > > > > > > >
> > > > > > > > > Hope this helps, and let us know if there is any further concern.
> > > > > > > > >
> > > > > > > > > Edit: We would like to add that for the maxpooling problem
> > > > > > > > > $$\\sup_{\\mathbf f \\in \\mathcal F} \max_{1\le i\le d,1\le j\le d'}(|\\mathbb E_p [\\mathcal A_q \\mathbf f]|)_{ij},$$
> > > > > > > > >
> > > > > > > > >  its optimal value equals $\max_i \mathbb E_p[|\nabla_i \log p - \nabla_i \log q|] $, and may have multiple maximizers, for example, any $\mathbf f$ that has a component equal to $sgn(\nabla_{i*} \log p - \nabla_{i*} \log q)$, where $i*=\arg\max_i  \mathbb E_p[|\nabla_i \log p - \nabla_i \log q|] $.

---

> ### Author Response · Authors · 2021-08-22
> **Thanks for your time and hope the response helps for your re-assessment of our work**
>
> Dear Reviewer c3DX,
>
> We sincerely hope our posted response can help for answering the questions. If you have any further comment or question, please let us know and we are glad to write a follow-up response.
>
> Thank you very much for the time reading the comments.

---

### Official Review · Reviewer_eewM · 2021-07-16

**Rating:** 7
**Confidence:** 3

**Summary:**

This work proposes to train two models jointly, one explicit density estimator, and one implicit generative model. The idea is that the explicit (energy) model specifies a valid and easy-to-evaluate likelihood function, while the implicit model is flexible enough to draw high-quality samples.
Because the explicit model is an energy model, we don't have the normalizing constant, meaning that we cannot evaluate how close the explicit model is to the implicit model. The authors propose to minimize the stein discrepancy between the implicit model and the energy model, and the Wasserstein distance between the implicit model and the data.

The contributions are proposed as follows:
* A joint learning framework from training an explicit energy model and an implicit model to learn to sample from a target distribution where we only have access to samples.
* A training algorithm that trains the implicit model to sample via minimizing the Wasserstein distance then trains the energy model to learn an explicit density by minimizing the stein discrepancy between the energy model and the implicit model.

**Limitations And Societal Impact:**

The authors mention that deep generative models can be used for generating false information, and therefore pose some risk.

**Main Review:**

I found this work enjoyable. The idea of learning a deep energy model via the stein discrepancy has been proposed before, but the addition of an implicit generative model really seems to help distribution fit. This is possibly due to the increased size of the generative model function space. [1, 2] consider the space of L2 functions, which need to vanish at the boundaries -- necessitating potentially harmful regularization on the generator function. This method avoids this issue by training an implicit model, that is known to perform well for sample generation.
Figure 4 and figure 5 were quite convincing to me. They show that both the implicit model and the energy model are able to capture the data distribution well, while their components trained independently more or less fail.
On a practical note, I believe it's clear from these toy examples that the learned distribution is more faithful with this joint training algorithm. But what is less clear is the advantage in high(er) dimensional inference tasks. The MNIST and CIFAR-10 results do not look much better than WGAN-GP. I see that the inception score is better for Joint-W, but the 2X training time cost is high given this margin.

As an aside, I found the fact that minimizing the stein discrepancy between the two models acts as a gradient penalty on the implicit model interesting. This might make sense since the energy model is "slower" than WGAN.


[1] Grathwohl, Will, et al. "Learning the stein discrepancy for training and evaluating energy-based models without sampling." International Conference on Machine Learning. PMLR, 2020.

[2] Hu, Tianyang, et al. "Stein neural sampler." arXiv preprint arXiv:1810.03545 (2018).

-----------------------
### Update After Rebuttal
After reading the other reviews/discussions, I agree that higher-dimensional inference tasks are important to evaluate on for this method. I do not intend to change my score, but I hope that the authors can show the applicability of their approach to tasks other than MNIST and CIFAR-10.


**Time Spent Reviewing:**

5

---

> ### Author Response · Authors · 2021-08-10
> **Response to Reviewer eewM**
>
> Thank you for summarizing our contribution precisely and highlights the technical aspects of the paper!
>
> We also thank the reviewer for comparing our work with [1-2] and shedding more lights on the merits of our approach, which we found insightful and will incorporate this discussion in our main text.
>
> As for the experiments on MNIST/CIFAR-10, indeed the improvement on sample generation is not much significant since we use standard architectures for two generative models (a shallow neural network). Further performance gain is expected when we use more complex instantiation of two models. For example, as further experiment, we change the specification of energy model to PixelCNN++ which is easy to control pixel-level features of generated images in order to add induction bias. In CIFAR-10, we manage to achieve inception score 7.25. Also, our results in Table 3 show that our model yields much better performance for out-of-distribution detection, which demonstrate superior density estimation than other explicit models in the high-dimensional inference tasks. A more distinctive advantage of our model other than the performance gains is that it enables sample generation and density estimation at one time in a unified framework. We believe that these improvements/advantages serve as more convincing merits of our model within reasonable training time.

---

> > ### Comment · Reviewer_eewM · 2021-08-30
> > **Thanks for your comments**
> >
> > I appreciate the authors' remarks with regard to my review.
> >
> > After reading the other reviews/discussions, I agree that higher-dimensional inference tasks are important to evaluate on for this method. I do not intend to change my score, but I hope that the authors can show the applicability of their approach to tasks other than MNIST and CIFAR-10.

---

> > > ### Author Response · Authors · 2021-08-30
> > > **Thank you for the feedbacks**
> > >
> > > Thanks for your feedbacks and comments. We agree that more complicated datasets would provide more evidence for supporting the effectiveness of our model. To this end, we also use LSUN dataset (a large-scale image dataset that is much more challenging than CIFAR-10) for our out-of-distribution detection task for testing the explicit model. Concretely, we treat CIFAR-10 as positive examples and LSUN images as negative examples. The model aims to distinguish the images from two datasets and is expected to give high density values for CIFAR-10 images and low density values for LSUN images. We compare our model with DEM, DGM and EGAN and obtain the results of Area Under of Curve (AUC): DEM 0.56, DGM 0.82, EGAN 0.56 and Joint-W 0.85. The results indicate that our model Joint-W can more accurately identify out-of-distribution samples from in-distribution ones, showing better empirical performance for density estimation. We believe that these results can further strengthen our contribution.
> > >
> > > We will evaluate on more datasets in future work, though the time limits for discussion period would make it hard for us to share more results at present.

---

### Author Response · Authors · 2021-08-10
**General Response by the Authors**

We thank the reviewers for their time, valuable feedback and constructive suggestions. Overall, the reviewers found our work interesting and novel (eewM, c3DX, p1LW), and appreciated our non-trivial theoretical results (c3DX), promising empirical performance (eewM, p1LW) and clear presentation (eewM, c3DX).

Before responding to each reviewer individually, we would like to restate our contribution in order to better resolve some big picture issues.

- **Methodology**
Our formulation involves three discrepancy measures that pursuit the consensus of two generative models (implicit and explicit) on closeness to data distribution, and we focus on Stein discrepancy that only requires unnormalized density from the explicit model and samples from the implicit model / real data. A comparison with recent works that attempt to unify both worlds is given in Table 1.

- **Theoretical analysis** By deriving closed-form expression for some infinite-dimensional optimization, our results in Section 3 show that the formulation enables mutual regularization effects for both models and can stabilize the training dynamics. The analysis are non-trivial, and to the best of our knowledge, analysis on the connection between Stein bridging term and regularization such as kernel Sobolev norm penalization are novel in the literature.

- **Empirical performance** The experiments in Section 4 further validate the framework by showing superior performance for both sample generation and density estimation over independently training a single model. We kindly refer to the Appendix for most experiment details.

In the following individual responses, we provide a thorough elaboration for all the raised issues and supplement new experiment results to further strengthen our contribution.

---

### Decision · Program_Chairs · 2021-09-27

**Decision:**

Accept (Poster)

**Comment:**

This paper proposes Stein bridge as a generic framework to jointly train an explicit generative model as well as an implicit model. Some theoretical analyses are provided. Experiments are conducted on synthetic data as well as image generation with MNIST data.

Reviewers agree that the proposed method is new. However, as cited by the authors, there are previous works on the joint training view of EBMs and implicit models. In this sense, the paper focuses on a known research topic but with a new methodology.

Reviewers agree that the theoretical analysis result is non-trivial so indeed a contribution. However, some clarity concerns remain, which needs to be improved in revision.

The major concern from the reviewers is that the experimental results are not strong, and certainly in GAN literature, MNIST is already a toy-ish dataset. They are not convinced that there is strong evidence that the demonstrated advantage on modelling corrupted data comes from the proposed Stein bridging method.